# Deuteration promotes circularly polarized light emission by suppression of vibration

Zhanxiang Chen ®[1], Manli Huang[1], Cheng Zhong[2], Mengcheng Wang[1], Jingsheng Miao[1] & Chuluo Yang ®[1] ✉

Circularly polarized light (CPL) is critical for advancing photonic technologies such as spin-based optical communication, quantum computing and displays. Developing these technologies necessitates CPL emitters with large dissymmetry factor ($g$) and high quantum yield ($\Phi$). However, there is an inherent trade-off between these two parameters. Here we integrate molecular deuteration into chiral thermally activated delayed fluorescence (TADF) emitters, leading to marked improvements in both $g$ and $\Phi$. Circularly polarized organic light-emitting diodes incorporating deuterated chiral TADF molecules as either emitters or host exhibit high performance, achieving maximum external quantum efficiency close to 40% and demonstrating up to over twofold enhanced electroluminescence $g$ compared to hydrogenated counterparts. Such advancements are attributed to suppression of vibrations by deuteration. Our deuteration strategy sets a foundation for designing organic chiral emitters with large $g$ and high $\Phi$, paving the way for high-performance CPL in display technologies.

Circularly polarized light (CPL) carrying spin angular momentum is essential for diverse applications such as chiral sensing, quantum computation, and display technologies[1]. However, materials capable of emitting CPL inevitably face considerable challenges in luminescence dissymmetry factor ($g$) and emission quantum yield ($\Phi$)[2,3]. This difficulty arises because $g$ is inversely proportional to the electric dipole transition moment ($\boldsymbol{\mu}$), while $\Phi$ is directly proportional to the square of $\boldsymbol{\mu}$, which is described as[4–7]

$$g \propto \frac{|\mathbf{m}| \cos\theta}{|\boldsymbol{\mu}|}, \qquad (1)$$

$$\Phi = \frac{k_r}{k_r + k_{nr}} \propto |\boldsymbol{\mu}|^2, \qquad (2)$$

where $\mathbf{m}$ is the magnetic dipole transition moment, $\theta$ is the angle between the $\mathbf{m}$ and $\boldsymbol{\mu}$ vectors, $k_r$ and $k_{nr}$ are the radiative transition and non-radiative transition rate constants, respectively. Generally,

achieving a high $\Phi$ requires a large $\boldsymbol{\mu}$, which in turn necessitates a larger $\mathbf{m}$ to maintain a large $g$. Given that $\mathbf{m}$-allowed transitions are typically $\boldsymbol{\mu}$-forbidden (or vice versa), it becomes difficult in chiral organics to achieve large $g$ while maintaining high $\Phi$. As such, this inherent tradeoff between $g$ and $\Phi$ poses significant limitations for the development of efficient CPL materials.

Recent empirical advances in CPL materials have involved in elucidating the intricate relationships between transition dipoles ($\mathbf{m}$ and $\boldsymbol{\mu}$) and their angle ($\theta$)[8–14] to explore solutions that enhance both $g$ and $\Phi$. Studies have shown that controlling molecular inter-segment twisting through spatial interactions[15,16] can lead to relatively high $g$ without compromising $\Phi$. However, this method is limited to specific CPL material systems and is not universally applicable, with some still displaying low $g$ (refs. [17–20]), low $\Phi$ (refs. [21–27]), or both[28–31], even after effectively reducing twisting vibrations. This has led to speculation that besides twisting vibrations, other molecular vibrations, notably carbon-hydrogen (C–H) stretching and C=C ring-stretching vibrations, might also impact $g$ and $\Phi$ (refs. [9,32,33]). Thus, the question

[1]Shenzhen Key Laboratory of New Display and Storage Materials, College of Materials Science and Engineering, Shenzhen University, Shenzhen, PR China. [2]Hubei Key Lab on Organic and Polymeric Optoelectronic Materials, Department of Chemistry, Wuhan University, Wuhan, PR China. ✉e-mail: clyang@szu.edu.cn

emerges: is it possible to reduce these molecular vibrations, thereby enhancing the $g$ and $\Phi$ of CPL materials concurrently?

Deuteration is a simple and effective method to decrease the amplitude of vibrational modes via isotopic effects from mass differences[33–36]. Additionally, it is known that the deuterium nucleus (deuteron[37]), comprising a proton and a neutron, alters molecular skeletal vibrational modes[35]. Given that changes in nuclear coordinates can influence the orbital contribution to $\mathbf{m}$ (which is positively correlated with $g$, Supplementary Note 1)[9,38], deuteration may have the potential to enhance $g$ in CPL materials. On the other hand, deuteration has been demonstrated to increase $\Phi$ (refs. 33,35,36,39). In this work, we design a deuterated CPL molecule and conceive that deuteration may enhance both the $g$ and $\Phi$ of CPL materials. Our results demonstrate a more than twofold increase in circular polarization ($g$-factor of $5.8 \pm 0.1 \times 10^{-3}$) and an enhanced $\Phi$ compared to the original hydrogenated counterpart. This performance boost is attributed to the deuteron's ability to suppress skeletal vibrations. Through the strategic adoption of a two-unit stacked tandem device configuration, circularly polarized (CP) OLEDs based on deuterated CPL materials as

chiral emitter achieve an electroluminescence dissymmetry factor ($g_{EL}$) of $6.4 \times 10^{-3}$ and an external quantum efficiency (EQE) of 39.3%. Additionally, the deuterated CPL material as chiral host results in a $g_{EL}$ of $4.6 \times 10^{-3}$ and an EQE of 37.7%. This work offers a universally applicable strategy for the development of high-performance CPL materials by simple deuteration.

## Results

The selection of R/S-CzCN as the template molecule was based on its structural rigidity, stemming from steric hindrance that limits structural twisting[40]. To systematically investigate the effect of molecular vibrations, deuterated and methyl-substituted derivatives of R/S-CzCN, namely R/S-D(16)CzCN and R/S-MeCzCN, were synthesized (Fig. 1a). The chemical structures of these enantiomers were fully characterized using $^1$H NMR, $^{13}$C NMR, high-resolution mass spectrometry, and single-crystal X-ray diffraction (Supplementary Figs. 1–9). Single-crystal X-ray diffraction analysis of R/S-D(16)CzCN (Supplementary Fig. 9) revealed slightly shortened C-D···π interactions (versus corresponding C-H···π interactions in R/S-CzCN) and increased crystal

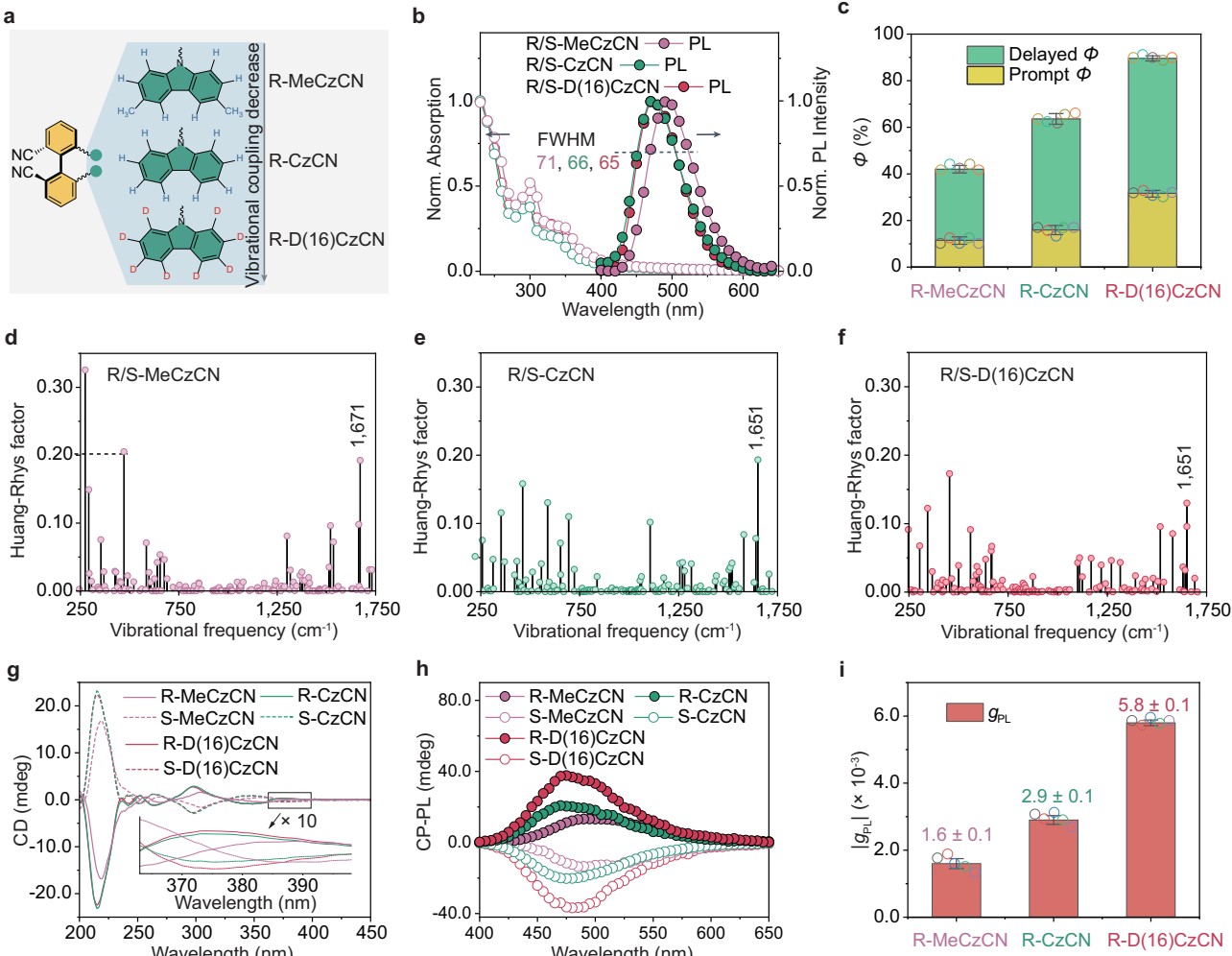

**Fig. 1 | Molecular design and characterization. a** Chemical structures of the designed CP-TADF emitters aimed at reducing molecular vibrations. **b** Absorption spectrum and PL spectra of R/S-MeCzCN, R/S-CzCN, and R/S-D(16)CzCN neat films at 298 K. **c** Prompt and delayed $\Phi$ values of R/S-MeCzCN, R/S-CzCN, and R/S-D(16)CzCN neat films. The data of prompt $\Phi$ and delayed $\Phi$ are represented as mean values ± standard deviation from the analysis of five individual samples, and the corresponding data points are overlaid. **d–f** Theoretically calculated Huang–Rhys factor for the $S_1 \rightarrow S_0$ electronic transitions plotted against the vibrational frequency

for R/S-MeCzCN (**d**), R/S-CzCN (**e**), and R/S-D(16)CzCN (**f**). **g** CD spectra of R/S-MeCzCN, R/S-CzCN, and R/S-D(16)CzCN neat films. **h** CP-PL spectra of R/S-MeCzCN, R/S-CzCN, and R/S-D(16)CzCN neat films. **i** $g_{PL}$ values of R/S-MeCzCN, R/S-CzCN, and R/S-D(16)CzCN neat films. The data of $g_{PL}$ is represented as mean values ± standard deviation from the analysis of five individual samples, and the corresponding data points are overlaid. Source data are provided as a Source Data file.

density, indicating that deuteration facilitates a denser crystal packing. Additionally, these enantiomers exhibited good thermal stability, with a decomposition temperature ($T_d$, at 5% weight loss) of approximately 340 °C (Supplementary Fig. 10).

The solid-state photophysical characteristics of the enantiomers R/S-MeCzCN, R/S-CzCN, and R/S-D(16)CzCN were investigated through ultraviolet-visible (UV–Vis) and photoluminescence (PL) spectroscopies. As shown in Fig. 1b, all enantiomers exhibited similar absorption profiles and broad emission bands centered around 470-490 nm. Notably, R/S-D(16)CzCN exhibited a narrower PL emission bandwidth than its non-deuterated analogs (R/S-MeCzCN, R/S-CzCN), a trend further confirmed in dilute solutions ($1 \times 10^{-5}$ M, 300 K; Supplementary Fig. 11). The PL decay curves (Supplementary Fig. 12) showed prompt/delayed lifetimes of 23.1 ns/4.7 μs for R/S-MeCzCN, 24.8 ns/4.2 μs for R/S-CzCN and 23.5 ns/4.5 μs for R/S-D(16)CzCN. A progressive increase in $\Phi$ was observed, starting from 42% for R/S-MeCzCN to 63% for R/S-CzCN (Supplementary Table 1) and reaching 90% for R/S-D(16)CzCN (Fig. 1c). Kinetic analysis (Supplementary Table 2 and Supplementary Note 2) indicated that the enhanced $\Phi$ in R/S-D(16)CzCN is due to a higher radiative decay rate constant ($1.4 \times 10^7$ s$^{-1}$) and a reduced non-radiative decay rate constant ($1.6 \times 10^6$ s$^{-1}$) relative to the counterparts, emphasizing the deuteration's role in boosting luminescence efficiency. We further performed the Molecular Materials Property Prediction Package (MOMAP) to explore the impact of deuteration on $\Phi$ through the Huang-Rhys factor, which quantifies the exciton-vibration coupling between excited ($S_1$) and ground ($S_0$) states (Fig. 1d–f). The calculations showed that the vibrational coupling was progressively reduced from R/S-MeCzCN (Supplementary Fig. 13)[41] to R/S-CzCN, then to R/S-D(16)CzCN, in line with the experimental observations. Specifically, deuteration significantly reduced the Huang-Rhys factor for the skeletal vibration mode (Supplementary Fig. 14), such as at 1651 cm$^{-1}$ in R/S-D(16)CzCN, thereby contributing to its enhanced $\Phi$.

We characterized the chiroptical properties of the CP-TADF emitters using both circular dichroism (CD) and circularly polarized photoluminescence (CP-PL) spectra. The CD spectra showed similar features, marked by alternating Cotton effects (Fig. 1g), with R/S-MeCzCN exhibiting a slight dip in intensity. We quantified the average absorption dissymmetry factor ($|g_{abs}|$) in the low-energy transition range (375–400 nm), with values of $1.2 \times 10^{-3}$ for R/S-MeCzCN, $1.5 \times 10^{-3}$ for R/S-CzCN, and $4.1 \times 10^{-3}$ for R/S-D(16)CzCN:

$$g_{abs} = \frac{2(A_{left} - A_{right})}{A_{left} + A_{right}} \tag{3}$$

where $A_{left}$ and $A_{right}$ are the absorption intensities of the left- and right-CPL, respectively (See Supplementary Fig. 15 for simulated CD spectra and Supplementary Table 3 for excited states; deuteration does not affect excitonic coupling). Importantly, the CP-PL spectra (Fig. 1h) revealed a distinct trend compared to the CD spectra: a reduction in molecular vibrations (Fig. 1a) corresponds to an increase in the absolute value of the photoluminescent dissymmetry factor ($|g_{PL}|$, Fig. 1i, Supplementary Figs. 16 and 17). Specifically, the average $|g_{PL}|$ (in the wavelength range of 430–580 nm) rose from $1.6 \pm 0.1 \times 10^{-3}$ for R/S-MeCzCN to $2.9 \pm 0.1 \times 10^{-3}$ for R/S-CzCN, and further to $5.8 \pm 0.1 \times 10^{-3}$ for R/S-D(16)CzCN. This represented a 3.6-fold increase compared to R/S-MeCzCN and a 2.0-fold increase compared to R/S-CzCN. Systematic CPL studies of all three CP-TADF enantiomer pairs in dilute solutions ($1 \times 10^{-5}$ M) at 300 K (Supplementary Fig. 11 and Supplementary Table 4) further confirmed that deuteration enhances the $g_{PL}$ of chiral emitters.

Theoretical analysis highlighted the impact of variations in nuclear mass among different isotopes. Conventional theoretical computations typically omit nuclear mass, which means isotope

effects do not alter the stationary-point structures (without considering vibrations) in this framework. To integrate deuteron's effects, we applied the Diagonal Born-Oppenheimer correction[42] to calculations, capturing the influence of nuclear mass. Our results demonstrated that the deuterated molecule R-D(16)CzCN exhibits larger magnetic dipole transition moment values (**m**, Supplementary Fig. 18) due to slight shifts in the biphenyl backbone nuclear coordinates upon deuteration (Supplementary Tables 5-8), but nearly identical electric dipole transition moments (**μ**) compared with its protonated isotopomer and the methyl-substituted counterpart (Fig. 2a). Of note, the incorporation of deuteron did not affect the angle ($\theta$) between the **m** and **μ** vectors (Supplementary Fig. 19). Therefore, the boost in **m** value directly enhanced the $g$, according to the relationship: $g \propto |\mathbf{m}|\cos\theta/|\mathbf{\mu}|$. However, the key point here was that the experimentally observed $g$ values of R-MeCzCN, R-CzCN and R-D(16)CzCN were not as large as those calculated based on stationary-point structures.

Considering that molecules are not static but influenced by vibrations (Fig. 2b), we performed first-principles vibrational analyses[43,44] for R-MeCzCN, R-CzCN, and R-D(16)CzCN. We computed the $g$ values for the normal modes associated with the $S_1 \rightarrow S_0$ electronic transition based on a simple Boltzmann model:

$$g_k = \frac{1}{2}\left(e^{-\Delta E_i/RT}g_i + e^{-\Delta E_j/RT}g_j\right), (i = +\delta u(k); j = -\delta u(k)) \tag{4}$$

$\Delta E_i$ and $\Delta E_j$ were the differences in excited state energies upon displacing the equilibrium $S_1$ geometry by a small dimensionless quantity ($+\delta u$, $-\delta u$) along the $k$th normal mode, within the harmonic limit, compared to the excited state energy at the equilibrium $S_1$ geometry. The dissymmetry factors $g_i$ and $g_j$ for these displaced structures were calculated through time-dependent density functional theory (TDDFT). Our calculations (Fig. 2c–e) indicated that by accounting for the Boltzmann distribution of vibrational mode energies, the overall average $g$-factor increases progressively from R-MeCzCN ($1.7 \times 10^{-3}$) to R-CzCN ($3.1 \times 10^{-3}$), and then to R-D(16)CzCN ($5.7 \times 10^{-3}$), which aligns with our experimental observations.

To better understand why these vibrational modes exhibited larger Boltzmann-weighted $g$-factors in R-D(16)CzCN relative to R-CzCN and R-MeCzCN, we computed the exciton wavefunction (transition density, $\rho$), which is critical for determining the $g$-factor. For these three compounds, the $S_1 \rightarrow S_0$ transition density was predominantly localized onto the biphenyl segment ($\rho$, Supplementary Fig. 20). According to previous systematic theoretical calculations[9,43], we recognized that molecualr vibrations significantly altered this localized transition density, extending beyond the biphenyl segment and thereby leading to a reduction in $g$-factor. Further analyses, therefore, focused on how this transition density alters with displacement along a normal mode, $\{\Delta\rho\}_\omega$ ($\omega$, frequency)[9,44]. For instance, Fig. 2f–h showed these differential wavefunction plots $\{\Delta\rho\}_\omega$, for a normal mode at approximately 1360 cm$^{-1}$, associated with C–C and C=C stretching vibrations of the chiral luminescent skeleton (Supplementary Fig. 21). In R-CzCN, perturbation along the 1356 cm$^{-1}$ mode altered the transition density within the carbazole-related C–H and nitrogen-centered regions, as well as the biphenyl C=C skeleton ($\{\Delta\rho\}_{1356}$, Fig. 2g). In R-MeCzCN, however, changes in the transition density not only involved the chiral luminescent skeleton but also extended to the alkyl CH$_3$ group ($\{\Delta\rho\}_{1366}$, Fig. 2f), where additional C-H bonds resulted in more pronounced changes to the $S_1$ transition density. In contrast, due to deuteration, R-D(16)CzCN showed suppressed transition density changes in these carbazole C–H and nitrogen-centered regions, as well as the carbazole C=C skeleton compared to R-CzCN ($\{\Delta\rho\}_{1356}$, Fig. 2h). Similar trends are observed for other representative vibrational modes, as shown in Supplementary Fig. 22. The pronounced transition density changes

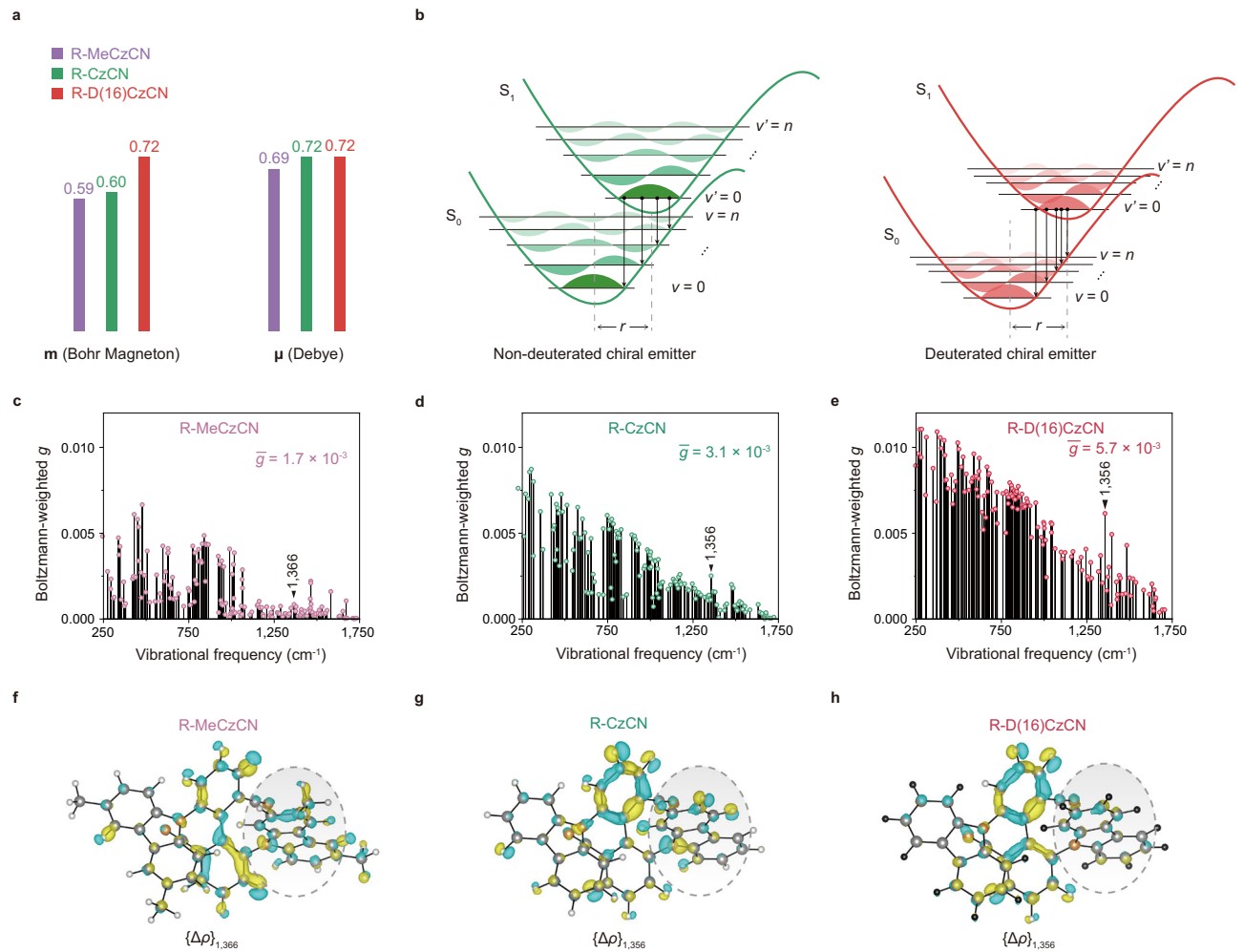

**Fig. 2 | Theoretical analysis. a** Representative magnetic (**m**) and electric (**μ**) dipole transition moments of three CP-TADF emitters: R-MeCzCN (violet), R-CzCN (green), and R-D(16)CzCN (red). **b** Schematic illustrating vibronic coupling between the $S_0$ and $S_1$ states of non-deuterated and deuterated chiral emitters. *r*, nuclear displacement; *υ* and *υ*′, vibrational quantum number of ground and excited state, respectively; *n*, vibrational quantum number of the *n*th vibrational state. Theoretically calculated Boltzmann-weighted *g* (from Eq. (4) using first-principles DFT and TDDFT) plotted against the vibrational frequency for R-MeCzCN (**c**); R-CzCN (**d**); R-D(16)CzCN (**e**). **f** Differential exciton wavefunction (transition density) upon displacement along the 1366 cm⁻¹ mode $\{\Delta\rho\}_{1{,}366\,\mathrm{cm}^{-1}}$, plotted for the $S_1 \rightarrow S_0$ transition of R-MeCzCN. Differential exciton wavefunction (transition density) upon displacement along the 1356 cm⁻¹ mode $\{\Delta\rho\}_{1{,}356\,\mathrm{cm}^{-1}}$, plotted for the $S_1 \rightarrow S_0$ transition of R-CzCN (**g**) and R-D(16)CzCN (**h**). Source data are provided as a Source Data file.

in R-MeCzCN and R-CzCN suggested that their transition densities are more susceptible to molecular vibrations, leading to significant *g*-factor losses compared to the *g*-values obtained from stationary-point TDDFT calculations (Supplementary Fig. 19). Conversely, since deuteration restricted skeletal vibrations (Fig. 1f), R-D(16)CzCN's transition density was less affected by molecular vibrations, resulting in smaller *g*-factor losses. In a word, deuteration in R-D(16) CzCN not only enhanced the **m** (Fig. 2a and Supplementary Fig. 23) but also suppressed skeletal vibrations, thereby achieving larger *g*-factor.

To translate the significant enhancements in *Φ* and $g_{PL}$ through deuteration into electroluminescence processes, we fabricated and investigated non-doped CP-OLEDs utilizing the three pairs of CP-TADF enantiomers as the chiral emitting layer (EML). This allowed us to further assess the impact of the deuterium substitution strategy on device performance. The optimized device structure (Fig. 3a) employed the following sequence: indium tin oxide (ITO)/1,4,5,8,9,11-hexaazatriphenylenehexacarbonitrile (HAT-CN, 5 nm)/1,1-bis[(di-4-tolylamino)phenyl]cyclohexane (TAPC, 30 nm)/tris(4-carbazolyl-9-ylphenyl)amine (TCTA, 15 nm)/3,3-di(9H-carbazol-9-yl)biphenyl (mCBP, 10 nm)/chiral EML (55 nm)/(1,3,5-triazine-2,4,6-triyl)

tris(benzene-3,1-diyl)tris(diphenylphosphine oxide) (POT2T, 20 nm)/1-(4-(10-([1,1′-biphenyl]-4-yl)anthracen-9-yl)phenyl)-2-ethyl-1H-benzo[d] imidazole (ANT-BIZ, 30 nm)/8-hydroxyquinolinato lithium (Liq, 2 nm)/ alumina (Al, 100 nm). In the device configuration, HATCN was used as a hole-injection layer; TAPC and TCTA functioned as the hole-transport layers, and ANT-BIZ was employed as electron-transport layers, respectively; mCBP and POT2T served as electron-blocking and hole-blocking layers to adjust the charge balance in the device; Liq and Al acted as the electron-injection layer and cathode, respectively. The chemical structures of the organic materials and the EL performance were depicted in Fig. 3b, c, Supplementary Fig. 24, and key parameters were provided in Table 1.

All the fabricated CP-OLEDs displayed nearly identical turn-on voltage of 2.8 V (Fig. 3b). The observed trends (Fig. 3c and Supplementary Fig. 25) in maximum external quantum efficiency (EQE) and electroluminescence dissymmetry factor ($g_{EL}$) values of these devices corresponded with the above mentioned *Φ* and $g_{PL}$ values (Fig. 1c, i). The device incorporating R-MeCzCN exhibited an EQE of $8.5 \pm 1.0\%$ and an average $g_{EL}$ of $1.9 \times 10^{-3}$. The device with R-CzCN as the emitter showed an EQE of $13.2 \pm 0.9\%$ and an average $g_{EL}$ of $3.2 \times 10^{-3}$. Remarkably, the device utilizing R-D(16)

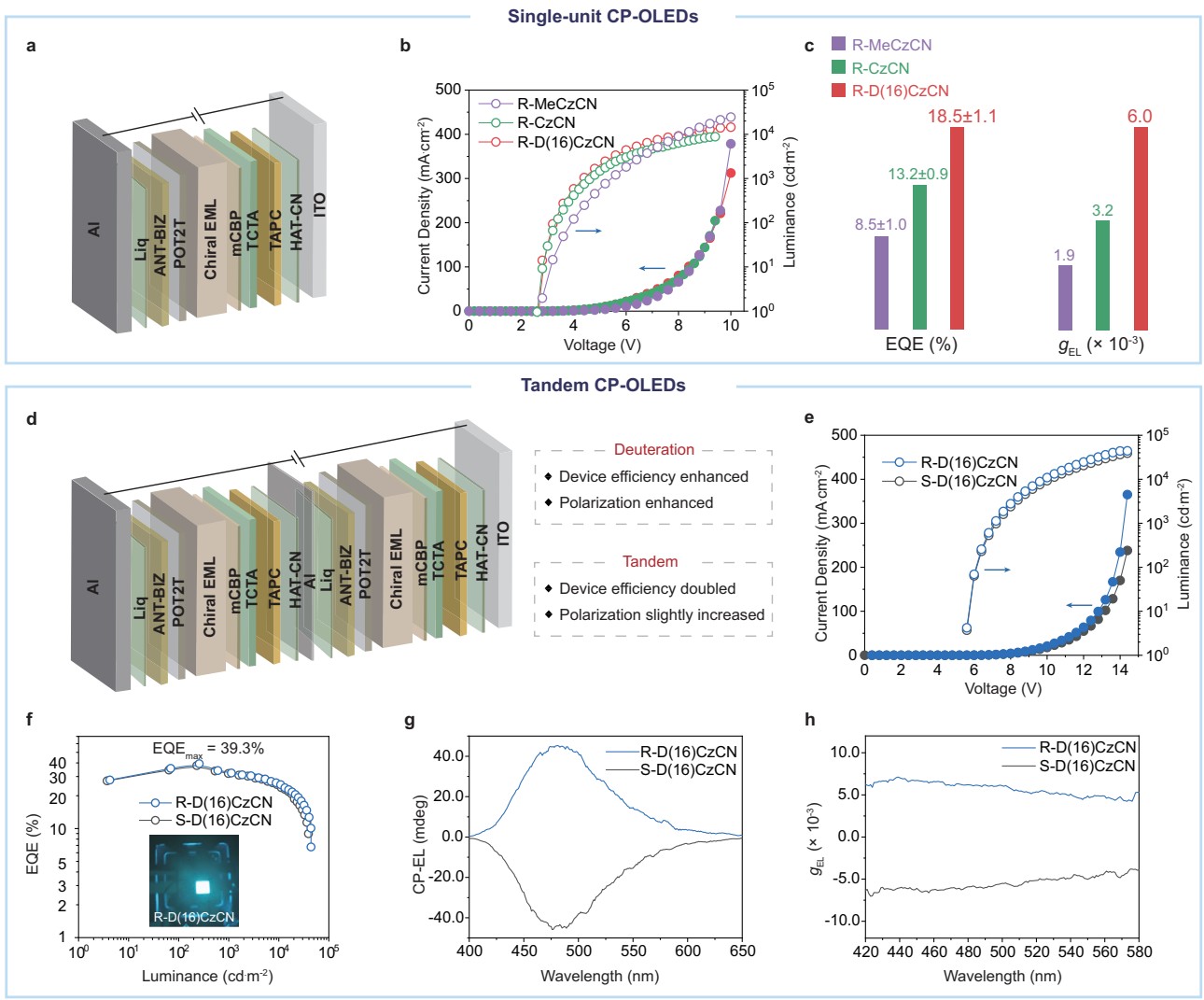

**Fig. 3 | Single-unit and tandem CP-OLEDs. a** Device structure of single-unit CP-OLEDs. Representative *J-V-L* curves (**b**), EQE$_{max}$, and $g_{EL}$ (**c**) of the single-unit CP-OLEDs utilizing three CP-TADF emitters: R-MeCzCN (violet), R-CzCN (green), and R-D(16)CzCN (red). **d** Device structure of tandem CP-OLEDs. Device performance of tandem CP-OLEDs based on R/S-D(16)CzCN: current density and luminance versus driving voltage curves (**e**); EQE versus luminance characteristics (**f**); CP-EL spectra (**g**); $g_{EL}$ values versus wavelength curves (**h**). Source data are provided as a Source Data file.

CzCN as the chiral emitter achieved the highest EQE of $18.5 \pm 1.1\%$ and an average $g_{EL}$ value of $6.0 \times 10^{-3}$, which is more than threefold that of R-MeCzCN-based devices and nearly twofold that of R-CzCN-based devices. These results clearly demonstrated the effectiveness of the deuterium substitution strategy in the electroluminescence

### Table 1 | The key device data for CP-OLEDs with the R configuration

| Chiral emitting layer | $V_{ON}$ (V) | CE (cd·A$^{-1}$) | PE (lm·W$^{-1}$) | EQE$^c$ (%) | $g_{EL}$ (×10$^{-3}$) |
|---|---|---|---|---|---|
| R-MeCzCN$^a$ | 2.8 | 21.7 ± 1.9 | 24.3 ± 2.8 | 8.5 ± 1.0(9.8) | +1.9 |
| R-CzCN$^a$ | 2.8 | 27.0 ± 1.9 | 30.3 ± 2.6 | 13.2 ± 0.9(14.9) | +3.2 |
| R-D(16)CzCN$^a$ | 2.8 | 38.2 ± 2.9 | 43.2 ± 3.5 | 18.5 ± 1.1(20.5) | +6.0 |
| R-D(16)CzCN$^b$ | 5.6 | 80.6 ± 3.6 | 39.7 ± 1.6 | 36.5 ± 1.7(39.3) | +6.4 |
| R-MeCzCN:BN2$^a$ | 2.8 | 74.2 ± 5.6 | 83.2 ± 6.3 | 17.9 ± 1.5(19.0) | +1.0 |
| R-CzCN:BN2$^a$ | 2.8 | 98.4 ± 4.4 | 110.4 ± 4.9 | 23.5 ± 1.3(26.3) | +1.9 |
| R-D(16)CzCN:BN2$^a$ | 2.8 | 152.4 ± 5.2 | 163.4 ± 10.7 | 36.3 ± 1.0(37.7) | +4.6 |

$^a$Average performance parameters for single-unit CP-OLEDs. $^b$Average performance parameters for tandem CP-OLEDs. $^c$The best EQE values are provided in parentheses. The mean values and standard deviations are based on measurements from 20 devices.

process, leading to simultaneous enhancements in both the EQE and $g_{EL}$.

In pursuit of enhancing the performance of chiral emitter-based CP-OLED, we fabricated a tandem architecture incorporating two chiral emissive layers, separated by a 1.5 nm Al interlayer (Fig. 3d and Supplementary Fig. 26). Demonstrated in Fig. 3e, the resulting tandem devices based on R-D(16)CzCN (depicted in blue) and S-D(16)CzCN (illustrated in black) achieved a peak luminance exceeding 44,000 cd·m$^{-2}$. These devices exhibited maximum EQE values of 39.3% for R-D(16)CzCN and 37.8% for S-D(16)CzCN (Fig. 3f), which are almost double compared to single-unit devices. At a practical luminance of 1000 cd·m$^{-2}$, the tandem devices sustained high EQE values of 32.7% for R-D(16)CzCN and 31.9% for S-D(16)CzCN. To evaluate the CP-EL characteristics of both tandem devices, we employed the JASCO CPL-300 under a voltage of 7 V. Figure 3g, h presented distinct CP-EL signals for both R-D(16)CzCN and S-D(16)CzCN-based tandem CP-OLEDs, with average $g_{EL}$ values recorded at $+6.4 \times 10^{-3}$ and $-6.3 \times 10^{-3}$, respectively, within the 420–580 nm wavelength range, indicating slight enhancement relative to single-unit devices.

To further evaluate the effectiveness of deuterium substitution in chiral host of CP-OLEDs, we utilized R/S-D(16)CzCN as the CP-

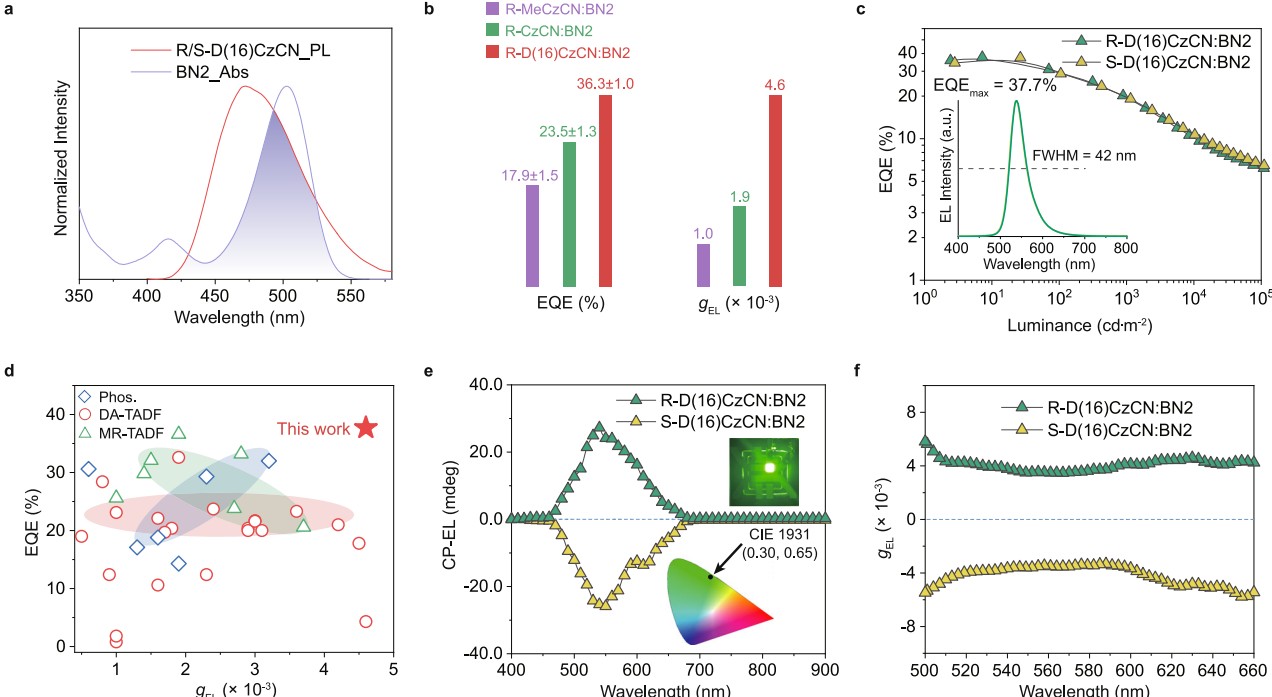

**Fig. 4 | CP-OLEDs based on deuterated CP-TADF hosts. a** Overlap between the emission spectrum of R/S-D(16)CzCN neat film and ultraviolet-visible absorption of BN2 in solution with a concentration of $10^{-5}$ mol·L$^{-1}$. **b** Representative EQE$_{max}$ and $g_{EL}$ of CP-OLEDs based on three CP-TADF hosts: R-MeCzCN:BN2 (violet), R-CzCN:BN2 (green), and R-D(16)CzCN:BN2 (red). **c** EQE versus luminance characteristics of CP-OLEDs based on R/S-D(16)CzCN:BN2. **d** The reported peak EQE and peak $g_{EL}$ values of various CP-OLEDs according to the data provided in Supplementary Table 9, measured using commercial CPL-200 or CPL-300 for CP-EL detection. Phos. (blue), DA-TADF (red) and MR-TADF (green) denote phosphorescent CP-OLED, CP-OLED utilizing donor-acceptor TADF emitter, and CP-OLED employing multi-resonance TADF emitter, respectively. CP-EL performance of CP-OLEDs based on R/S-D(16)CzCN:BN2: CP-EL spectra and CIE chromaticity diagram (inset) (**e**); $g_{EL}$ values versus wavelength curves (**f**). Source data are provided as a Source Data file.

TADF host and selected BN2[45], an achiral MR-TADF emitter with strong diastereomeric interaction[46,47] (for details regarding the selection of achiral guests, see Supplementary Figs. 27–35 and Supplementary Note 3), as the guest emitter. Co-deposited R/S-D(16) CzCN:BN2 films displayed an additional CD signal in the 400–500 nm range, which differed from that of pristine R/S-D(16)CzCN neat films (Supplementary Fig. 28), suggesting induced chirality in BN2[48]. Furthermore, the absorption spectrum of BN2 substantially overlapped with the emission spectrum of the R/S-D(16)CzCN (Fig. 4a), and the CPL spectra of R/S-D(16)CzCN:BN2 films varied systematically with the excitation wavelength (290–410 nm, Supplementary Fig. 28), ensuring efficient energy transfer in the fabricated CP-OLEDs. These CP-OLEDs were constructed using a single-unit device structure: ITO/ HAT-CN (5 nm)/TAPC (30 nm)/TCTA (15 nm)/mCBP (10 nm)/EML (2 wt% BN2 in R/S-D(16)CzCN, 55 nm)/POT2T (20 nm)/ANT-BIZ (30 nm)/Liq (2 nm)/Al (100 nm). For comparison, control CP-OLEDs based on R/S-MeCzCN:BN2 and R/S-CzCN:BN2 were also fabricated with the same optimized doping concentration of 2 wt%. All CP-OLEDs initiated operation at a consistent turn-on voltage of 2.8 V and featured narrowband electroluminescence spectra with peaks at 538 nm, full-width at half-maximum values of 42 nm, confirming emission completely from BN2. This suggested effective exciton formation on the CP-TADF hosts with subsequent Förster resonance energy transfer to BN2 (Langevin recombination). The CP-OLEDs incorporating these three pairs of CP-TADF hosts exhibited consistent performance trends (Fig. 4b) with devices using CP-TADF molecules as emitters alone (Fig. 3c). The R-D(16)CzCN:BN2-based CP-OLED achieved an EQE$_{max}$ of 37.7% (Fig. 4c), surpassing the R-CzCN:BN2-based device by 40% and doubling the efficiency of the R-MeCzCN:BN2-based counterparts (Table 1, Supplementary

Figs. 36 and 37). We highlighted that the efficiency of the R-D(16) CzCN:BN2-based device was the highest among the reported CP-OLEDs (Fig. 4d and Supplementary Table 9). At a voltage of 5 V, CP-EL measurements obtained with a JASCO CPL-300 spectrometer showed distinct circular polarization. The R-D(16)CzCN:BN2-based devices exhibited a $g_{EL}$ of $4.6 \times 10^{-3}$, exceeding the $1.9 \times 10^{-3}$ and $1.0 \times 10^{-3}$ of the devices based on R-CzCN:BN2 and R-MeCzCN:BN2, respectively (Fig. 4e, f; Supplementary Fig. 38 and Supplementary Note 4). Moreover, operational stability assessments at an initial luminance of 1000 cd·m$^{-2}$ indicated an LT$_{80}$ (time for luminance to decay to 80% of its initial value) of 106 h for the R-D(16)CzCN:BN2-based CP-OLED, significantly longer than the 37 and 24 h achieved by the R-CzCN:BN2 and R-MeCzCN:BN2-based devices (Supplementary Fig. 39). Collectively, these results unequivocally demonstrated that deuteration within chiral host molecules significantly bolsters the overall performance of CP-OLEDs, in terms of efficiency, circular polarization, and operational stability.

To demonstrate the generality of our deuteration strategy for CPL enhancement, we designed another structurally rigid deuterated CPL molecule, (P,P)/(M,M)-D(32)CzTBCO (Fig. 5 and Supplementary Figs. 40–42). As anticipated, this deuterated CPL material exhibited significantly enhanced |$g_{PL}$| values across various dilute solutions (Supplementary Fig. 43 and Supplementary Table 10). Furthermore, when employed as chiral emitters in CP-OLEDs (Fig. 5a), the devices based on (P,P)/(M,M)-D(32)CzTBCO (Fig. 5b) achieved an EQE of 18.9% and an average |$g_{EL}$| value of $1.0 \times 10^{-2}$ in the wavelength range between 480 and 600 nm (Fig. 5c, d, Supplementary Figs. 44 and 45). Compared with the hydrogenated counterpart (Supplementary Fig. 46), the EQE and |$g_{EL}$| values were increased by 1.5-fold and 1.3-fold, respectively.

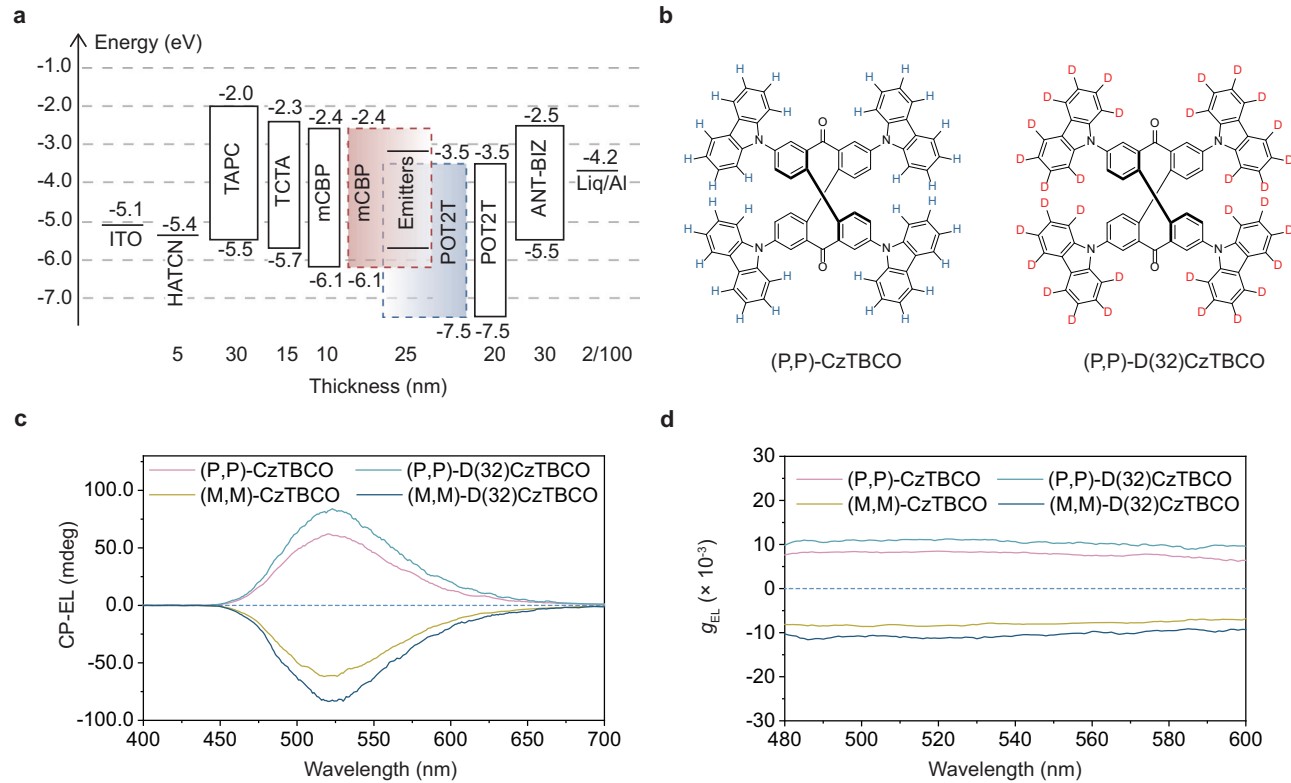

**Fig. 5 | Another example demonstrating the effectiveness of the deuteration strategy. a** Device structure of CP-OLEDs using (P,P)/(M,M)-CzTBCO and (P,P)/(M,M)-D(32)CzTBCO as the chiral emitters. **b** Chemical structures of (P,P)-CzTBCO and (P,P)-D(32)CzTBCO. CP-EL performance of CP-OLEDs based on (P,P)/(M,M)-CzTBCO and (P,P)/(M,M)-D(32)CzTBCO as the chiral emitters: CP-EL spectra (**c**); $g_{EL}$ values versus wavelength curves (**d**). Source data are provided as a Source Data file.

## Discussion

Taken together, our experiments and simulations demonstrate that deuteration concurrently enhances both $g_{PL}$ and $\Phi$ of CPL material. CP-OLEDs employing deuterated CP-TADF molecules, as either emitters or host, not only attain markedly higher efficiencies but also exhibit enhanced $g_{EL}$ values compared to their hydrogenated counterparts. Our findings suggest that deuteration is a strategy to minimize the influence of vibrational modes that reduce the CPL signal. For chiral molecules that are minimally affected by molecular vibrations, deuteration may little affect on $g_{PL}$ (Supplementary Fig. 47 and Supplementary Table 11). This has important implications for the design of organic chiral emitters for CP-OLED technology. The proposed design principles may also open up new possibilities for spin electronic devices, where deuterated organic chiral materials could potentially enhance spin-related phenomena such as chiral-induced spin selectivity[1,49].

## Methods

### Theoretical calculations

All DFT calculations were performed using the Gaussian-16 package[50]. The DFT-D3(BJ) method was applied to include the empirical dispersion correction[51]. The Huang-Rhys factors were determined using the Molecular Materials Property Prediction Package (MOMAP) 2022 A package[52]. Geometric optimization with the Diagonal Born-Oppenheimer correction was performed using Coupled-Cluster techniques for Computational Chemistry (CFOUR)[53] at the CCSD(T)/cc-pVTZ level[42]. To compute the vibrational properties of excited states and the effect of vibrations on the dissymmetry factor, we coupled our molecular TDDFT calculations to finite displacement methods[43]. TDDFT calculations were performed at the CAM-B3LYP[54]/def2-SVP[55] level.

## Materials

All reagents and solvents were purchased from commercial sources and used without any additional purification unless specifically mentioned. 3,6-dimethyl-9H-carbazole, sodium hydride, and dehydrated DMF were purchased from Energy Chemical Technology (Shanghai) Co., Ltd. 9H-carbazole-1,2,3,4,5,6,7,8-d8, 6,6'-difluoro-[1,1'-biphenyl]-2,2'-dicarbonitrile, R/S-6,6'-di(9H-carbazol-9-yl)-[1,1'-biphenyl]-2,2'-dicarbonitrile (R/S-CzCN, Supplementary Fig. 48) and (P,P)/(M,M)-2,7,11,16-tetra(9H-carbazol-9-yl)tetrabenzo[a,c,f,h][10]annulene-9,18-dione ((P,P)/(M,M)-CzTBCO, Supplementary Fig. 49) were synthesized and purified according to the literature[36,40,56]. Thin layer chromatography employed glass 0.25 mm silica gel plates. Flash chromatography columns were packed with 200–300 mesh silica gel in petroleum ether (bp. 60–90 °C). HPLC chromatograms were obtained on an Agilent 1100 series system using a Chiralpak AD column with hexane/ethanol (60/40) or hexane/ethyl acetate (75/25) as the eluent at a flow rate of 1.0 mL·min⁻¹. Optical rotations (Supplementary Table 12) were measured using a 1 mL cell with a 1 dm path length on a Rudolph Autopol I polarimeter at 589 nm.

### Materials synthesis

**Preparation of R/S-D(16)CzCN.** Under an argon atmosphere, 9H-carbazole-1,2,3,4,5,6,7,8-d8 (1.75 g, 10.00 mmol) and sodium hydride (0.40 g, 16.70 mmol) were combined with dehydrated DMF (30 mL) and stirred at room temperature for 30 min. Following this, 6,6'-difluoro-[1,1'-biphenyl]-2,2'-dicarbonitrile (1.20 g, 5.00 mmol) was added to the reaction mixture, and the resulting solution was stirred at 80 °C for 12 h. Upon cooling to room temperature, the black mixture was diluted with water and extracted with dichloromethane (3 × 30 mL). The organic phases were dried over anhydrous sodium sulfate and the solvent was removed under reduced pressure. After

purification by column chromatography on silica gel using dichloromethane/petroleum ether (v/v = 1/1) as eluent, the racemic mixture (1.20 g, yield: 45%) was obtained as a white solid. Following further purification by recrystallization from a chloroform/hexane mixture, the racemic mixture yielded R-D(16)CzCN and S-D(16)CzCN enantiomers (Supplementary Fig. 50) via chiral high-performance liquid chromatography (HPLC). The absolute configuration was subsequently confirmed through crystallographic analysis. Finally, R-D(16)CzCN and S-D(16)CzCN were separately purified through sublimation twice under reduced pressure for CP-OLED fabrication. $^1$H NMR (CDCl$_3$, 400 MHz) δ (ppm): 7.33 (d, $J$ = 8.0 Hz, 2H), 7.49 (t, $J$ = 8.0 Hz, 2H), 7.98 (d, $J$ = 8.0 Hz, 2H). $^{13}$C NMR (CDCl$_3$, 100 MHz) δ (ppm): 108.4, 108.7, 108.9, 109.1, 109.4, 117.5, 118.1, 118.7, 118.9, 119.2, 119.6, 119.8, 120.0, 120.3, 123.8, 123.9, 125.0, 125.2, 125.4, 125.5, 125.8, 131.3, 133.4, 134.5, 136.9, 138.7, 140.2, 141.4. HRMS (ESI): m/z [M + H]$^+$ calcd for C$_{38}$H$_7$D$_{16}$N$_4$$^+$: 551.2921; found: 551.2904. CCDC numbers: 2267192 and 2285327.

**Preparation of R/S-MeCzCN.** The title compound was synthesized according to the same procedure with the R/S-D(16)CzCN except to use 3,6-dimethyl-9H-carbazole (1.95 g, 10.00 mmol) to replace 9H-carbazole-1,2,3,4,5,6,7,8-d8. After purification by column chromatography on silica gel using dichloromethane/petroleum ether (v/v = 1/1) as eluent, the racemic mixture (1.33 g, yield: 45%) was obtained as a white solid. Following further purification by recrystallization from a chloroform/hexane mixture, the racemic mixture yielded R-MeCzCN and S-MeCzCN enantiomers (Supplementary Fig. 51) via chiral HPLC. The absolute configuration was subsequently confirmed through crystallographic analysis. Finally, R-MeCzCN and S-MeCzCN were separately purified through sublimation twice under reduced pressure for CP-OLED fabrication. $^1$H NMR (CDCl$_3$, 400 MHz) δ (ppm): 2.36 (s, 6H), 2.45 (s, 6H), 5.39 (d, $J$ = 8.0 Hz, 2H), 6.28 (d, $J$ = 8.0 Hz, 2H), 7.00 (d, $J$ = 8.0 Hz, 2H), 7.06 (d, $J$ = 8.0 Hz, 2H), 7.28 (d, $J$ = 8.0 Hz, 2H), 7.44-7.48 (m, 4H), 7.67 (s, 2H), 7.94 (d, $J$ = 8.0 Hz, 2H). $^{13}$C NMR (CDCl$_3$, 100 MHz) δ (ppm): 21.3, 29.7, 30.2, 31.5, 109.2(2), 117.4, 118.1, 119.0, 120.1, 126.8, 127.1, 129.4, 129.6, 131.2, 133.2, 134.4, 137.0, 139.0, 139.1, 140.2. HRMS (ESI): m/z [M + H]$^+$ calcd for C$_{42}$H$_{31}$N$_4$$^+$: 591.2543; found: 551.2542. CCDC numbers: 2280086 and 2280087.

**Preparation of (P,P)/(M,M)-D(32)CzTBCO.** Under an argon atmosphere, 2,7,11,16-tetraiodotetrabenzo[a,c,f,h][10]annulene-9,18-dione (0.66 g, 0.76 mmol), 9H-carbazole-1,2,3,4,5,6,7,8-d8 (1.07 g, 6.11 mmol), tri-tert-butylphosphonium tetrafluoroborate (0.04 g, 0.15 mmol), tris(dibenzylideneacetone)dipalladium(0)-chloroform complex (0.08 g, 0.08 mmol) and sodium t-butoxide (1.17 g, 12.22 mmol) were combined with toluene (30 mL) and stirred at 120 °C for 24 h. Upon cooling to room temperature, the black mixture was diluted with water and extracted with dichloromethane (3 × 30 mL). The organic phases were dried over anhydrous sodium sulfate and the solvent was removed under reduced pressure. After purification by column chromatography on silica gel using dichloromethane/petroleum ether (v/v = 1/1) as eluent, the racemic mixture (0.36 g, yield: 45%) was obtained as a yellow solid. Following further purification by recrystallization from a chloroform/hexane mixture, the racemic mixture yielded (P,P)-D(32)CzTBCO and (M,M)-D(32)CzTBCO enantiomers (Supplementary Fig. 52) via chiral HPLC. The absolute configuration was subsequently confirmed through crystallographic analysis. Finally, (P,P)-D(32)CzTBCO and (M,M)-D(32)CzTBCO were separately purified through sublimation twice under reduced pressure for CP-OLED fabrication. $^1$H NMR (CDCl$_3$, 500 MHz) δ (ppm): 8.17 (d, $J$ = 2.5 Hz, 4H), 7.68 (dd, $J_1$ = 8.0 Hz, $J_2$ = 2.5 Hz, 4H), 7.41 (d, $J$ = 8.0 Hz, 4H). $^{13}$C NMR (CDCl$_3$, 125 MHz) δ (ppm): 195.1, 139.9, 139.5, 138.7, 138.2, 132.7, 129.4, 127.7, 125.7, 123.7, 120.2, 120.1, 109.2. HRMS (ESI): m/z [M]$^+$ calcd for C$_{74}$H$_{12}$D$_{32}$N$_4$O$_2$: 1052.5478; found: 1052.5488.

**NMR.** $^1$H NMR and $^{13}$C NMR were measured on Bruker Advanced II (400 MHz or 500 MHz) spectrometer. m, s, and d correspond to the $^1$H NMR spectra with multiple peaks, single peak, and double peaks, respectively. The chemical shifts (δ) were given in parts per million (ppm) related to internal tetramethylsilane (TMS, 0.00 ppm for $^1$H NMR), CDCl$_3$ (77.00 ppm for $^{13}$C NMR).

**HRMS.** High-resolution mass spectra (HRMS) were measured on Thermo Scientific LTQ Orbitrap XL Gas Chromatography Mass spectrometer.

**XRD.** X-ray diffractometer (XRD) analysis was conducted on a Bruker APEX-II CCD diffractometer using a Cu Kα radiation source (λ = 1.54178 Å). The crystal packing structures were solved and refined with the Bruker SHELXTL-2018 software package.

**TGA.** Thermogravimetric analysis (TGA) measurements were performed on the Rigaku TG/DTA8122 instrument under a nitrogen atmosphere. The thermal stability of the powdered samples was determined by measuring their weight loss, heated at a rate of 10 °C·min$^{-1}$ from room temperature to 600 °C. The decomposition temperature ($T_d$) was determined as the temperature that corresponds to a 5% weight loss of the initial weight of the compounds.

**CV.** Cyclic voltammetry (CV) measurements were conducted under a nitrogen atmosphere using a CHI 660 A Electrochemical Workstation with Ferrocecne/Ferrocenium (Fc/Fc$^+$) used as the calibration standard. The supporting electrolyte consisted of 0.1 M tetra-n-butylammonium hexafluorophosphate (Bu$_4$NPF$_6$) in either anhydrous dichloromethane (for HOMO measurement) or tetrahydrofuran (for LUMO measurement) solution. To carry out the measurements, a platinum plate, a platinum wire, and a saturated Ag/AgCl electrode were employed as a working electrode, a counter electrode, and a reference electrode, respectively. To determine the HOMO and LUMO energy levels, the following equations were used:

$$HOMO = - \left( E_{ox} - E_{(Fc/Fc^+)} + 4.8 \right) eV \quad (5)$$

$$LUMO = - \left( E_{red} - E_{(Fc/Fc^+)} + 4.8 \right) eV \quad (6)$$

where $E_{ox}$ and $E_{red}$ are the onset oxidation potential and onset reduction potential relative to Ag/AgCl, respectively, and $E_{(Fc/Fc^+)}$ is the onset oxidation potential of the ferrocene external standard.

**Fabrication of thin films.** Quartz substrates were sequentially cleaned by ultrasonic treatment in acetone and ethanol, followed by drying under a nitrogen gas flow. The substrates were then treated for 10 min using a UV-ozone surface processor (PL16 series, Sen Lights Corporation). Organic thin films for optical measurements were deposited onto the cleaned quartz substrates via thermal evaporation under high vacuum conditions (≤ 5.0×10$^{-5}$ Pa).

**UV–Vis absorption spectra.** Ultraviolet-visible (UV–Vis) absorption spectra were acquired using a Shimadzu UV-2700 recording spectrophotometer with baseline correction. The sample under analysis was an undoped thin film of the material, obtained by depositing organic materials on a quartz glass substrate using a vacuum evaporation technique. The film samples had a thickness of 25 nm.

**PL characterization.** Photoluminescence (PL) spectra were recorded using a Hitachi F-4600 fluorescence spectrophotometer. The transient PL decay curves were measured via FluoTime 300 (PicoQuant GmbH) utilizing a Picosecond Pulsed UV-LASER (LASER375) as the excitation source. The film samples had a thickness of 100 nm.

**Quantum yields**. Absolute photoluminescence quantum yields ($\Phi$) were obtained using a Hamamatsu UV-NIR absolute PL quantum yield spectrometer (C13534, Hamamatsu Photonics), equipped with a calibrated integrating sphere. All film samples were excited at 250 nm, and during the $\Phi$ measurements, the integrating sphere was purged with pure and dry nitrogen to maintain an inert environment.

**CD spectra**. The electronic circular dichroism (CD) spectra were obtained utilizing a J-1500 CD spectrometer (JASCO) at a scanning rate of 100 nm·min$^{-1}$, over the range of 200–700 nm, with baseline correction applied.

**CP-PL spectra**. Circularly polarized photoluminescence (CP-PL) spectra were recorded using a CPL-300 spectrophotometer (JASCO) within the 400–600 nm range, with an excitation wavelength of 330 nm for all film samples. The CP-PL spectra were subsequently analyzed using JASCO's Spectra Manager software (Version 2.12.00) to extract the corresponding $g_{PL}$ spectra.

**CP-OLED fabrication and characterization**. All organic materials used in the experiments, with the exception of the chiral emitting layer, were sourced from Lumtec, Inc. Prior to use, all compounds were subjected to temperature-gradient sublimation under high vacuum. The CP-OLEDs were fabricated on ITO-coated glass substrates with multiple organic layers sandwiched between the transparent bottom ITO anode and the top metal cathode. The ITO-coated glass was cleaned ultrasonically and sequentially with acetone and ethanol, followed by drying with nitrogen gas flow and treatment with a UV-ozone surface processor PL16 series (Sen Lights Corporation). All material layers were then deposited through vacuum evaporation in a vacuum chamber under a high vacuum ($\leq 5.0 \times 10^{-5}$ Pa). The deposition system permitted the complete device structure to be fabricated in a single vacuum pump-down without breaking the vacuum. The deposition rate for the various layers of the CP-OLEDs was controlled, with the hole injection layer (HATCN), the hole-transporting layers (TAPC, TCTA, and mCBP), and the electron-transporting layers (POT2T and ANT-BIZ) all being deposited at a rate of 0.2-0.3 nm·s$^{-1}$. The chiral emitting layer was prepared by evaporating CP-TADF emitters from the individual source while controlling the rate at 0.1 nm·s$^{-1}$. And the doping was conducted by co-evaporation of the CP-TADF host and MR-TADF emitters from two evaporation sources with different evaporation rates. The electron-injection layer (Liq) and aluminum cathode were deposited sequentially, with rates of 0.03 and 0.5 nm·s$^{-1}$, respectively. To remove oxygen and moisture, the CP-OLEDs with an active area of 0.09 cm$^2$ were finally encapsulated in glass caps. The electroluminescence characteristics were performed using a Keithley 2400 direct-current source meter and an absolute EQE measurement system (C9920-12, Hamamatsu Photonics, Japan). The average external quantum efficiency, current efficiency, and power efficiency of the devices were calculated based on 20 measured devices. CP-EL spectra were measured on a CPL-300 spectrophotometer (JASCO) with "Standard" sensitivity at 200 nm·min$^{-1}$ scan speed and a response time of 2.0 s employing "band" mode.

**Stability test**. The stability test of the encapsulated CP-OLEDs was conducted using a luminance meter (FS-MP64, Suzhou FSTAR Scientific Instrument Co. Ltd., China) under constant current density driving conditions with an initial luminance of 1000 cd·m$^{-2}$.

## Data availability

Crystallographic data for the structures reported in this article have been deposited at the Cambridge Crystallographic Data Centre (CCDC), under deposition numbers CCDC 2267192 (R-D(16)CzCN), 2285327 (S-D(16)CzCN), 2280086 (R-MeCzCN) and 2280087 (S-MeCzCN). Copies of the data can be obtained free of charge via https://

www.ccdc.cam.ac.uk/structures/. The full set of data generated in this study has been deposited in the Figshare database under accession code https://doi.org/10.6084/m9.figshare.26631226. Source data are provided with this paper.

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

## Acknowledgements

This work was funded by the National Natural Science Foundation of China (grant no. 52403237 to Z.C.; grant no. 52130308 to C.Y.), the

Shenzhen Science and Technology Program (grant no. JCYJ20220818095816036 to C.Y.; grant no. RCBS20231211090518026 to Z.C.), Guangdong Basic and Applied Basic Research Foundation (grant no. 2022A1515110445 to Z.C.) and Scientific Foundation for Youth Scholars of Shenzhen University (grant no. 868-000001033371 to Z.C.). We thank X. Yin and X. Cao (College of Materials Science and Engineering, Shenzhen University) for their assistance with the synthesis of (P,P)/(M,M)-CzTBCO materials and the paper writing, respectively. We thank the Instrumental Analysis Center of Shenzhen University for the assistance with the Nuclear Magnetic Resonance spectroscope technical support.

## Author contributions

C.Y. supervised the projects. Z.C. designed and conducted the experiments and performed theoretical calculations. M.W. synthesized the (P,P)/(M,M)-CzTBCO and (P,P)/(M,M)-D(32)CzTBCO materials. Z.C., M.H., C.Z., J.M., and C.Y. participated in discussions. Z.C., M.H., and C.Y. contributed to the manuscript writing. All authors engaged in result analysis and provided feedback on the manuscript.

## Competing interests

SZU has filed patent applications on materials and devices. C.Y., Z.C., M.H., and J.M. are the authors of the invention. CN patent application no. 2024119477798 (pending). The other authors declare no competing interests.
