## [Transparent Peer Review file · Nature Communications]

Deuteration promotes circularly polarized light emission by suppression of vibration

Corresponding Author: Professor Chuluo Yang

Version 0:

Reviewer comments:

Reviewer #1

(Remarks to the Author)

This manuscript describes the synthesis, study, and application of CP-TADF emitters in OLEDs, derived from a molecular design published by the Chen group in *Angewandte Chemie International Edition* a few years ago. (ref. 51, which probably deserves more prominent acknowledgment in the manuscript, Axially Chiral TADF-Active Enantiomers Designed for Efficient Blue Circularly Polarized Electroluminescence. *Angew. Chem. Int. Ed.* 59, 3500-3504 (2020)).

To summarize this contribution, the authors synthesize a deuterated version of this CP-TADF material and compare its properties with those of non-deuterated and methylated analogues. Based on their data, the authors propose that deuteration can be considered "a universally applicable strategy for the development of high-performance CPL materials." To explain the enhancement of the g factor, the authors perform advanced theoretical calculations (note that the reviewer does not have the expertise to evaluate this part). The authors then apply this material as a chiral dopant or chiral host to construct efficient CP-OLED devices (showing also the known effect of perdeuteration on device lifetimes and EQE).

Although this work shows considerable promise, additional experiments and major revisions are necessary before I can recommend its publication in *Nature Communications*.

1) The authors should measure the CPL of all three emitters (R/S-MeCzCN, R/S-CzCN, and R/S-D(16)CzCN) in solution and compare with data provided in ref. 51. Is the g-factor enhancement only measured in the solid state? If yes, the authors should explain why. A study of CPL in various solvents (hexane, toluene, DCM, and methanol) at 10⁻⁵ M concentration for both deuterated and hydrogenated materials is also recommended.

2) Please provide HPLC chromatograms for all compounds used in the study.

3) In ref. 51, the PLQY of S-CzCN in thin film is reported as 40%. In this study, it is measured around 20% for PF and 60% with DF. Please explain this discrepancy or verify the measurements.

4) Does the deuteration affect the geometry (e.g., dihedral angles, C-H- π , C-D- π distances, etc.) of Cz-Ax-CN (named here CzCN)? (The crystal structures of the non-deuterated material are described in ref. 51.)

5) The gEL of the CP-OLED constructed by the authors using R-CzCN (3.2×10^{-3}) is 4-5 times lower than the gEL of the CP-OLED described in ref. 51 (1.4×10^{-2}). Even the deuterated material led to a gEL more than 2 times lower than the CP-OLED device described in ref. 51. I understand that it is not the same OLED stack, but can the authors provide an explanation beyond that? Where is the point corresponding to the results reported in ref 51 in figure 4d ?

6) Please provide gPL spectra of the EML used in CP-OLEDs using both hydrogenated and deuterated materials (host + chiral dopant (deuterated and non-deuterated) and chiral host (deuterated and non-deuterated) + achiral MR-TADF).

7) The authors should also describe in more detail the mechanisms and rationale behind the circularly polarized light (CPL) emission of the OLED involving R/S-D(16)CzCN as host.

8) Finally, the author's state: "This work offers a universally applicable strategy for the development of high-performance

CPL materials by simple deuteration." In my opinion, to substantiate this claim, the authors should demonstrate that the CPL enhancement is valid for at least one other example of a CPL-active molecule (at least in solution or thin film state).

Reviewer #2

(Remarks to the Author)

Chen et al. introduced a novel approach for constructing circularly polarized organic light-emitting diodes (CP-OLEDs) by incorporating hydrogen isotopes into their chiral emitters. Their work combines two previously established methods: deuterated TADF emitters and chiral TADF materials based on twisted backbones, which make the LUMO regions chiral, to achieve high-efficiency CP-OLEDs. While the authors present impressive results regarding efficiency and other performance metrics, I do not believe this manuscript is suitable for publication in Nature Communications, which prioritizes highly original scientific contributions.

The primary method employed, deuteration of chiral TADF, is essentially a combination of two well-established techniques. Neither the deuteration process nor the dissymmetrization of the TADF structure represents a novel contribution by the authors. While it is true that groundbreaking results can sometimes arise from the combination of existing methods, this manuscript does not seem to fit that paradigm. Deuterated TADF emitters have been explored as early as 2021 (see <https://onlinelibrary.wiley.com/doi/full/10.1002/smcs.202000057>), and the efficiency of these systems has steadily improved, with some reaching external quantum efficiencies (EQE) of over 30% (e.g., <https://onlinelibrary.wiley.com/doi/full/10.1002/adom.202300981> and <https://onlinelibrary.wiley.com/doi/10.1002/adom.202401391>).

It is unclear whether the authors intend to emphasize the circularly polarized (CP) aspect or the OLED efficiency in this work, as their title suggests the former, but the content does not strongly support either focus. They base their system on the R/S-CzCN compound, which was first reported in *Angewandte Chemie* (<https://onlinelibrary.wiley.com/doi/full/10.1002/anie.201914249>). It is problematic that this critical reference is only cited in 'method-related reference [ref 51]', rather than being discussed in the main body of the text. Moreover, the original publication reports much higher g_{EL} values (-1.2×10^{-2} and $+1.4 \times 10^{-2}$) than those presented in this manuscript, which are 20-40 times lower. This significant discrepancy is not addressed in the current submission.

While I commend the authors for achieving efficiencies as high as 30% or even approaching 40%, the low g_{EL} values raise concerns if the paper's goal is to present new chiral emission mechanisms. Comparing their deuterated molecule to the non-deuterated version from the *Angewandte* work, it seems likely that deuteration has significantly diminished the chiroptical response (from 1.2×10^{-2} to $+6.4 \times 10^{-3}$). The authors might be better served by highlighting the non-chiral performance and optimization strategies for high-efficiency OLEDs, as the current results suggest this work aligns more with "high-efficiency OLEDs with weak circular polarization."

In the context of chiral OLEDs and circularly polarized luminescence (CPL), the community is in need of systems that push g_{EL} values to 10^{-2} or higher. The benchmark for g_{EL} has been set at approximately 10^{-3} since 2015, particularly with EQE>30% systems that can be achieved by simply separating commercially available emitters (<https://www.nature.com/articles/srep14912>).

Reviewer #3

(Remarks to the Author)

The authors describe the design of chiral emitters displaying circularly polarized thermally activated delayed fluorescence (CP-TADF) and explore how deuteration of the donor unit can potentially impact the photophysical and chiroptical properties of these emitters. They compare the optoelectronic properties of three different systems and conclude that deuteration of the emitters leads to an increase in both the photoluminescence quantum yield (PLQY) and the intensity of circularly polarized luminescence (CPL). They also demonstrate this beneficial effect in CP-OLEDs. While deuteration of organic compounds has already been explored with similar effects on PLQY, the observed increase in CPL intensity (by a factor of 2) is quite novel, in my opinion, and should be of interest to researchers in this field.

Although I believe this manuscript could be suitable for publication in Nature Communications, I am curious about how generalizable this strategy is, which I consider a crucial point for Nature research journals. Additionally, I would appreciate seeing more photophysical and chiroptical data recorded in solution to better understand this "deuterium effect."

Here are some comments that may be of interest to the authors:

Overall, I feel that certain turns of phrase used in the manuscript could be revised for clarity. Moreover, the cited publications on CPL intensity in organic systems are heavily focused on CP-TADF emitters. While I understand this focus, I believe that papers describing CPL emitters with g_{EL} values around $\sim 10^{-2}$ also merit citation. For example: P. Reine et al., Chiral double-stapled o-OPEs with intense circularly polarized luminescence, *Chem. Commun. (Camb)*, 2019, 55, 10685; K. Dhbaibi et al., Achieving high circularly polarized luminescence with push-pull helicenic systems: from rationalized design to top-emission CP-OLED applications, *Chem. Sci.*, 2021, 12, 5522.

Have the authors explored the solvatochromism of these CPL emitters using solvents with different polarities, for both luminescence and CPL measurements?

I find the connection between references 9 and 36 and the sentence, "Additionally, it is known that the deuterium nucleus

(deuteron), comprising a proton and a neutron, has a ground-state total spin of 1 (ref. 35). Given that the spin contribution to m (which is positively correlated with g)... unclear. Could the authors provide more details here?

Have the authors measured the g_{abs} for the low-energy transition of their compounds? It seems that the sign of the latter is reversed when comparing ECD and CPL (i.e., the R enantiomers give positive CPL but negative ECD). Am I correct? Could the authors clarify this point?

How were the films prepared?

The authors mention that the radiative decay becomes faster for deuterated emitters while the non-radiative decay decreases, yet the emitters exhibit the same luminescence lifetime. Why is this the case?

The reduction in molecular vibrations in the luminescence spectra is not entirely clear to me. Would it be useful to analyze the full width at half maximum (FWHM)?

The Huang-Rhys analysis is quite interesting. However, CzCn and MeCzCn display similar patterns despite differences in PLQY. How can these results be interpreted?

The authors mention the CISS effect in the conclusion, but no experiments related to this phenomenon are presented. What exactly are the authors suggesting here? This is not clear to me.

Version 1:

Reviewer comments:

Reviewer #1

(Remarks to the Author)

The authors have significantly improved their manuscript in light of the reviewers' comments. While the pronounced effect of deuteration on the dissymmetry factors observed for CzCN (a twofold enhancement) remains surprising to me, the additional data provided—as well as the demonstration that the glum enhancement effect is also observable with another pair of CP-TADF emitters (1.3 fold)—are convincing.

Before I can recommend the publication of this valuable contribution in Nature Communications, I have some final requests.

1) On page 8, lines 119–120, the authors mention that they measured g_{abs} values in the range of the S0-S1 transitions approximately ten times lower than $glum$ values. This seems counterintuitive, unless there is a significant geometrical difference between the ground and excited states. Could the authors clarify or provide a rationale for this observation?

2) Regarding the CPL data particularly when comparing the hydrogenated and deuterated CP-TADF emitters, providing the raw data ($\Delta I = f(\lambda)$ curves and the $DC = f(\lambda)$) would be helpful.

Reviewer #2

(Remarks to the Author)

The authors have made a substantial effort in revising the manuscript by including a significant number of new experimental results, such as additional solution-state CPL measurements and another example of a chiral emitter. I appreciate their responsiveness to the reviewers' concerns and the work invested in this revision. However, despite these additions, I regret to say that I still cannot recommend this manuscript for publication in Nature Communications.

While the manuscript now includes more data, the core scientific claims remain insufficiently supported. Many of the conclusions are based on speculative reasoning without direct experimental or theoretical validation. In several instances, key arguments rely heavily on external references, some of which contain controversial or unverified claims within the chiral optoelectronics community. This reliance undermines the manuscript's overall scientific robustness.

That said, it is encouraging to see that all three reviewers have highlighted the previously hidden citation of Ref. 51 in the Methods section. It is appropriate that the authors have now integrated this important reference into the main text to properly acknowledge foundational contributions from the community.

Specific comments:

The authors report an increase in g_{PL} from 2.9×10^{-3} to 5.8×10^{-3} via deuteration, attributing the enhancement to increased magnetic transition dipole moments (m). However, from Supplementary Fig. S17, there does not appear to be a significant contribution in density distributions from the deuterium atoms. It would strengthen the manuscript if the authors could clarify whether and how these atoms significantly affect the frontier molecular orbitals. If their contribution is indeed minor, it is not possible to attribute such m changes to the deuteration directly and requires better justification.

The discussion around so-called "chiral FRET" lacks both theoretical grounding and experimental support. The authors cite a 2013 *Angewandte Chemie* paper as the primary justification, yet that study itself does not offer direct evidence for a chiral FRET mechanism—only a coexistence of CPL and FRET processes. Without detailed calculations or mechanistic insights, it is difficult to accept the implication that a non-chiral emitter in a chiral host can emit CPL at the reported levels. I would consider it is just a result of host material self-absorption or selective scattering, not necessary to be bound with other mechanisms. Known literature suggests that CPL emission typically requires chirality in the emitter or structural changes in the excited state—conditions not addressed here or in those systems that time-reversal symmetry is broken. David L Andrews' classic paper (<https://iopscience.iop.org/article/10.1088/2050-6120/ab10f0>) clearly demonstrates that achiral

emitters cannot generate CPL solely through energy transfer in a chiral host, unless additional symmetry-breaking mechanisms are involved.

If the authors were indeed able to induce CPL in an achiral emitter solely via chiral FRET without invoking such mechanisms, this would constitute a remarkable discovery with implications beyond CPL enhancement, potentially touching on fundamental aspects of quantum electrodynamics or CPT theorem. However, the manuscript does not currently present sufficient evidence to support such a claim.

Conclusion:

Although the manuscript presents some decent OLED performance data (without any doubts), its core contributions to the understanding of molecular chirality and CPL mechanisms remain speculative and, in some cases, potentially misleading. The logical framework often relies on secondary sources without critical evaluation, and the foundational claims require more rigorous theoretical and experimental substantiation.

I therefore do not support the publication of this manuscript in its current form.

Reviewer #3

(Remarks to the Author)

Dear authors,

Thanks for having addressed most of the concerns raised by the referees. I would be happy to recommend your work for publication.

A minor concern for me relies on the exact relation between the deuteration and the increase of ϕ_{CPL} . While I trust your results, I am just wondering why other chiral deuterated compounds have not shown this effect. It would be nice to provide a comparison to me.

Version 2:

Reviewer comments:

Reviewer #2

(Remarks to the Author)

All concerns have been addressed by the authors and I have no more comments.

Reviewer #3

(Remarks to the Author)

The authors have attempted to address the concerns raised by the referees. While I acknowledged their revisions last time, I am now less convinced with their additional experimental work.

Indeed, it now seems that the effect of the deuteration is more related to the dihedral angle within the biaryl electron acceptor bridge than to its involvement in the frontier molecular orbital. Accordingly, this is more of a steric effect that could eventually be obtained with another atom rather than a specific deuterium, if I am correct. In my opinion, the title should be changed in order to align more closely with the experimental results.

The authors tried to respond to referee 1 but their response is far from being clear. They invoked an instrumental error in treating the data, and then used calculations to ascertain their "new" results. Based on their measurements, I am therefore also wondering if the effect of the deuteration may impact the potential presence of excitonic coupling within these chiral emitters. This would ultimately have a direct consequence on the chiroptical properties obtained in both the absorption and emission processes. Have the authors taken this into consideration?

I also have some concerns about the potential FRET mechanism. Why should there be a dependence on excitation wavelengths between 290 and 330 nm? The chiral donor should only transfer energy from its lowest excited state, if I am correct. Explanations based on ECD responses (Fig. S27, for example) are not relevant to me because the very low ECD values may lead to errors in interpretation.

Version 3:

Reviewer comments:

Reviewer #3

(Remarks to the Author)

I would like to thank the authors for this work.

All concerns have been addressed by the authors and I have no more comments.

Dear Editor and Reviewers,

Enclosed please find our revised manuscript entitled “**Deuteration promotes circularly polarized light emission**” (Manuscript ID: **NCOMMS-24-50744A**) for publication as a research article in *Nature Communications*. We sincerely thank the reviewers for their insightful and professional comments, which helped us substantially improve the quality of our manuscript.

We revised the manuscript thoroughly in response to the reviewers’ comments and the point-by-point responses are provided. The major improvements upon revision include:

1. The generality of the deuteration strategy is demonstrated through the design and synthesis of another deuterated chiral emitters, (P,P)/(M,M)-D(32)CzTBCO, which enabled CP-OLEDs to achieve a 1.3-fold enhancement of average $|g_{EL}|$ value (1.0×10^{-2}) and high device efficiency compared to the hydrogenated analogue.
2. Solution-phase PL and CPL studies are conducted across various solvents (hexane, toluene, dichloromethane, and methanol) at 10^{-5} M concentration.
3. The crystallographic, photophysical, and chiroptical data are deeply analyzed, and detailed mechanistic discussions were added in the supplementary information.
4. The HPLC chromatogram test is performed and included in the supplementary information.
5. The figures in the manuscript are optimized for enhanced clarity.

For your reference, annotated files of the revised manuscript and supporting information with all changes highlighted in yellow are also attached. We hope that the revised paper could now be satisfied for your requirements.

Yours Sincerely,

Chuluo Yang on behalf of the authors.

Corresponding author: **Professor Chuluo Yang**

at College of Materials Sciences and Engineering, Shenzhen University, Shenzhen 518071, People’s Republic of China. Tel: +86-0755-86713985, e-mail: clyang@szu.edu.cn.

We greatly appreciate the editor and reviewers for taking the time to review and critique our manuscript. Accordingly, we have fully addressed the concerns raised by the reviewers.

Reviewer #1:

General comment: *This manuscript describes the synthesis, study, and application of CP-TADF emitters in OLEDs, derived from a molecular design published by the Chen group in *Angewandte Chemie International Edition* a few years ago. (ref. 51, which probably deserves more prominent acknowledgment in the manuscript, *Axially Chiral TADF-Active Enantiomers Designed for Efficient Blue Circularly Polarized Electroluminescence. Angew. Chem. Int. Ed.* 59, 3500-3504 (2020)).*

To summarize this contribution, the authors synthesize a deuterated version of this CP-TADF material and compare its properties with those of non-deuterated and methylated analogues. Based on their data, the authors propose that deuteration can be considered "a universally applicable strategy for the development of high-performance CPL materials." To explain the enhancement of the g factor, the authors perform advanced theoretical calculations (note that the reviewer does not have the expertise to evaluate this part). The authors then apply this material as a chiral dopant or chiral host to construct efficient CP-OLED devices (showing also the known effect of perdeuteration on device lifetimes and EQE).

*Although this work shows considerable promise, additional experiments and major revisions are necessary before I can recommend its publication in *Nature Communications*.*

Response: We sincerely appreciate the reviewer's positive and professional comments. In fact, the molecular design reported by the Chen group in *Angewandte Chemie International Edition* (originally ref. 51, now ref. 40 in the revised manuscript) was explicitly acknowledged and served as the primary inspiration for the development of our CP-TADF emitters.

Comment 1: *The authors should measure the CPL of all three emitters (R/S-MeCzCN, R/S-CzCN, and R/S-D(16)CzCN) in solution and compare with data provided in ref. 51. Is the g-factor enhancement only measured in the solid state? If yes, the authors should explain why. A study of CPL in various solvents (hexane, toluene, DCM, and methanol) at 10^{-5} M concentration for both deuterated and hydrogenated materials is also recommended.*

Response: We thank the reviewer for the constructive suggestion. In adherence to this guidance, we have conducted solution-phase CPL studies for all three CP-TADF enantiomer pairs (R/S-MeCzCN, R/S-CzCN, and R/S-D(16)CzCN) in four solvents (hexane, toluene, DCM, methanol) at 10^{-5} M concentration. These results are now presented in **Supplementary Fig. 11** and **Supplementary Table 3**. The solution data reveal the same trend as observed in the solid state: the average $|g_{\text{PL}}|$ values (in the wavelength range of 430-580 nm) increase progressively from $1.8 \pm 0.1 \times 10^{-3}$ for R/S-MeCzCN to $3.2 \pm 0.1 \times 10^{-3}$ for R/S-CzCN, and further

to $5.8 \pm 0.2 \times 10^{-3}$ for R/S-D(16)CzCN. These findings confirm that deuteration effectively enhances the g_{PL} factor in dilute solutions.

Additional sentence, (manuscript, page 8): “Systematic CPL studies of all three CP-TADF enantiomer pairs in dilute solutions (1×10^{-5} M) at 300 K (Supplementary Fig. 11 and Supplementary Table 3) further confirmed that deuteration enhances the g_{PL} of chiral emitters.”

Supplementary Fig. 11. Photophysical and chiroptical properties of R/S-MeCzCN, R/S-CzCN, and R/S-D(16)CzCN in different solvents (1×10^{-5} M) at 300 K. (a) PL spectra of R/S-MeCzCN in different solvents (1×10^{-5} M) at 300 K. (b) PL spectra of R/S-CzCN in different solvents (1×10^{-5} M) at 300 K. (c) PL spectra of R/S-D(16)CzCN in different solvents (1×10^{-5} M) at 300 K. (d) CP-PL spectra of R/S-MeCzCN in different solvents (1×10^{-5} M) at 300 K. (e) CP-PL spectra of R/S-CzCN in different solvents (1×10^{-5} M) at 300 K. (f) CP-PL spectra of R/S-D(16)CzCN in different solvents (1×10^{-5} M) at 300 K. (g) g_{PL} values versus wavelength curves of R/S-MeCzCN in different solvents (1×10^{-5} M) at 300 K. (h) g_{PL} values versus wavelength curves

of R/S-CzCN in different solvents (1×10^{-5} M) at 300 K. (i) g_{PL} values versus wavelength curves of R/S-D(16)CzCN in different solvents (1×10^{-5} M) at 300 K.

Supplementary Table 3. Maximum and average g_{PL} values for R/S-MeCzCN, R/S-CzCN, and R/S-D(16)CzCN across various solvents, measured within the 430-580 nm spectral range.

Molecule	Solvent	Max g_{PL}^a	Average g_{PL}^a	Max g_{PL}^b	Average g_{PL}^b
R/S-MeCzCN	hexane	2.9×10^{-3}	1.5×10^{-3}	-2.8×10^{-3}	-1.8×10^{-3}
	toluene	2.2×10^{-3}	1.7×10^{-3}	-2.5×10^{-3}	-1.9×10^{-3}
	dichloromethane	2.3×10^{-3}	1.7×10^{-3}	-2.9×10^{-3}	-1.9×10^{-3}
	methanol	4.3×10^{-3}	1.8×10^{-3}	-3.6×10^{-3}	-1.7×10^{-3}
R/S-CzCN	hexane	4.2×10^{-3}	3.3×10^{-3}	-4.5×10^{-3}	-3.1×10^{-3}
	toluene	4.3×10^{-3}	3.3×10^{-3}	-4.8×10^{-3}	-3.3×10^{-3}
	dichloromethane	4.1×10^{-3}	3.3×10^{-3}	-4.0×10^{-3}	-3.3×10^{-3}
	methanol	3.9×10^{-3}	2.9×10^{-3}	-4.2×10^{-3}	-3.1×10^{-3}
R/S-D(16)CzCN	hexane	7.5×10^{-3}	6.0×10^{-3}	-8.0×10^{-3}	-5.8×10^{-3}
	toluene	7.1×10^{-3}	6.1×10^{-3}	-7.3×10^{-3}	-5.8×10^{-3}
	dichloromethane	8.5×10^{-3}	5.9×10^{-3}	-8.0×10^{-3}	-6.1×10^{-3}
	methanol	7.1×10^{-3}	5.4×10^{-3}	-1.0×10^{-2}	-5.5×10^{-3}

^aMaximum g_{PL} values and average g_{PL} values for the R-enantiomer. ^bMaximum g_{PL} values and average g_{PL} values for the S-enantiomer.

Comment 2: Please provide HPLC chromatograms for all compounds used in the study.

Response: We appreciate the reviewer for the suggestion. Accordingly, we have included the HPLC chromatograms for all synthesized compounds. These details are now comprehensively presented in the Supplementary Information, specifically within **Supplementary Figs. 35-39**.

Comment 3: In ref. 51, the PLQY of S-CzCN in thin film is reported as 40%. In this study, it is measured around 20% for PF and 60% with DF. Please explain this discrepancy or verify the measurements.

Response: We thank the reviewer for the astute observation and valuable suggestion. Upon examining the reference, we found that the reported PLQY value of 40% for S-CzCN in thin film does not clearly specify the measurement conditions, such as whether it was conducted in air or under vacuum, nor does it detail the film preparation method (solution-processed or vacuum-deposited) or the excitation wavelength used. To verify our measurements, we re-prepared R/S-CzCN thin films using vacuum deposition and systematically remeasured their PLQYs under both air and vacuum conditions using varied excitation wavelengths at room temperature. The corresponding data are provided in **Supplementary Table 1**.

Supplementary Table 1. Absolute Φ values of R/S-CzCN neat films measured in air and under vacuum conditions at room temperature with different excitation wavelengths.

	250 nm	270 nm	290 nm	310 nm	330 nm	350 nm	370 nm
Air	61.2%	60.9%	57.2%	54.3%	50.5%	49.2%	46.5%
Vacuum	62.8%	62.2%	60.6%	58.9%	56.5%	55.8%	54.3%

Comment 4: Does the deuteration affect the geometry (e.g., dihedral angles, C-H- π , C-D- π distances, etc.) of Cz-Ax-CN (named here CzCN)? (The crystal structures of the non-deuterated material are described in ref. 51.)

Response: We appreciate the reviewer for the professional comment. In response, we have carefully compared the crystal structures of the deuterated (R/S-D(16)CzCN) and non-deuterated (R/S-CzCN) compounds. As shown in **Supplementary Fig. 7**, deuteration has minimal influence on the intramolecular geometry, including the dihedral angles within the molecular framework. Additionally, the intermolecular C \equiv N $\cdots\pi$ distances remain essentially unchanged (**Supplementary Fig. 9**). However, we observed a slight but measurable shortening of the intermolecular C-D $\cdots\pi$ interaction relative to the corresponding C-H $\cdots\pi$ interactions (2.842 vs. 2.863 Å for R-enantiomers; 2.841 vs. 2.865 Å for S-enantiomers, **Supplementary Fig. 9**). This reduction of approximately 0.021-0.024 Å is consistent with the increased crystal densities of R/S-D(16)CzCN (1.287–1.288 g cm⁻³) compared to their non-deuterated counterparts (1.284 g cm⁻³). Thus, deuteration promotes a more compact and denser crystal packing environment.

Additional sentence, (manuscript, page 4): “Single-crystal X-ray diffraction analysis of R/S-D(16)CzCN (**Supplementary Fig. 9**) revealed slightly shortened C-D $\cdots\pi$ interactions (versus corresponding C-H $\cdots\pi$ interactions in R/S-CzCN) and increased crystal density, indicating that deuteration facilitates a denser crystal packing.”

Supplementary Fig. 7. Single crystal structures of R/S-D(16)CzCN. (a) R-D(16)CzCN. (b) S-D(16)CzCN. The two crystals were grown by slow evaporating a solution in CH₂Cl₂:methanol (5:2) at 25°C. Selected bond

length (Å) and angles (°): For R-D(16)CzCN: N1-C13 1.418(2) Å, N4-C21 1.422(3) Å, C14-C20 1.492(3) Å, \angle C7-N1-C13-C18 -42.9(3)°, \angle C27-N4-C21-C22 -52.3(3)°, \angle C13-C14-C20-C21 -62.1(3)°; For S-D(16)CzCN: N7-C14 1.418(2) Å, C26-N28 1.422(2) Å, C19-C21 1.492(2) Å, \angle C6-N7-C14-C15 42.8(2)°, \angle C29-N28-C26-C25 52.3(2)°, \angle C14-C19-C21-C26 62.3(2)°. For comparison, crystallographic data of R/S-CzCN (CCDC codes: 1945359 for R-CzCN, 1945360 for S-CzCN) were obtained from the Cambridge Crystallographic Data Centre. Selected bond length (Å) and angles (°): For R-CzCN: N1-C25 1.421 Å, N2-C32 1.421 Å, C30-C31 1.493 Å, \angle C12-N1-C25-C26 -42.8(2)°, \angle C37-N3-C21-C22 -52.1(7)°, \angle C25-C30-C31-C32 -62.2(4)°; For S-CzCN: N1-C25 1.421 Å, N2-C33 1.421 Å, C30-C32 1.493 Å, \angle C24-N2-C33-C34 42.8(8)°, \angle C1-N1-C25-C26 52.2(0)°, \angle C25-C30-C32-C33 62.2(1)°.

Supplementary Fig. 9. Single-crystal X-ray diffraction analysis of R/S-D(16)CzCN. (a) R-D(16)CzCN. (b) S-D(16)CzCN. Selected crystallographic data: For R-D(16)CzCN: C-D... π 2.842 Å, C \equiv N... π 2.984 Å, C \equiv N... π 3.243 Å, crystal density: 1.287 g cm⁻³; For S-D(16)CzCN: C-D... π 2.841 Å, C \equiv N... π 2.982 Å, C \equiv N... π

3.244 Å, crystal density: 1.288 g cm⁻³. For comparison, crystallographic data of R/S-CzCN (CCDC codes: 1945359 for R-CzCN, 1945360 for S-CzCN) were obtained from the Cambridge Crystallographic Data Centre. Selected crystallographic data: For R-CzCN, C-H...π 2.863 Å, C≡N...π 2.981 Å, C≡N...π 3.248 Å, crystal density: 1.284 g cm⁻³; For S-CzCN, C-H...π 2.865 Å, C≡N...π 2.983 Å, C≡N...π 3.247 Å, crystal density: 1.284 g cm⁻³.

Comment 5: *The g_{EL} of the CP-OLED constructed by the authors using R-CzCN (3.2×10^{-3}) is 4-5 times lower than the g_{EL} of the CP-OLED described in ref. 51 (1.4×10^{-2}). Even the deuterated material led to a g_{EL} more than 2 times lower than the CP-OLED device described in ref. 51. I understand that it is not the same OLED stack, but can the authors provide an explanation beyond that? Where is the point corresponding to the results reported in ref 51 in figure 4d?*

Response: We appreciate the reviewer for the insightful inquiry. The apparent discrepancy in g_{EL} values arises from differences in measurement methodologies: ref. 40 (originally ref. 51) employed a quarter-wave plate/linear polarizer system, which is prone to optical artifacts that may artificially inflate g_{PL} or g_{EL} values, as corroborated by recent studies (*Nat. Commun.* 14, 1065 (2023); *Adv. Mater.* 35, 2302279 (2023)). In contrast, all g_{EL} values in our study were measured consistently using the JASCO CPL-300 spectrometer, which is the commercial instrument recognized for reliable and reproducible CP-EL measurements. Due to differences in measurement methodologies and the potential artifacts involved, direct comparison between our results and those in ref. 40 would not be scientifically meaningful. Consequently, to maintain data consistency and ensure reliable comparison, the data point from ref. 40 was intentionally excluded from Fig. 4d, which exclusively summarizes values obtained with commercial CPL-200 or CPL-300 instrument.

Modified sentence, (manuscript, page 14): "...Supplementary Table 2, **measured using commercial CPL-200 or CPL-300 for CP-EL detection**"

Comment 6: *Please provide g_{PL} spectra of the EML used in CP-OLEDs using both hydrogenated and deuterated materials (host + chiral dopant (deuterated and non-deuterated) and chiral host (deuterated and non-deuterated) + achiral MR-TADF).*

Response: We appreciate the reviewer's suggestion. Accordingly, we have provided the g_{PL} spectra of the EMLs employed in the CP-OLEDs, utilizing deuterated (R/S-D(16)CzCN) and non-deuterated (R/S-CzCN) materials as chiral emitters or chiral hosts. These data are now presented in **Supplementary Figs. 15 and 40**.

Supplementary Fig. 15. g_{PL} values versus wavelength curves of deuterated R/S-D(16)CzCN and non-deuterated R/S-CzCN as chiral emitters.

Supplementary Fig. 40. g_{PL} values versus wavelength curves of deuterated R/S-D(16)CzCN and non-deuterated R/S-CzCN as chiral hosts.

Comment 7: *The authors should also describe in more detail the mechanisms and rationale behind the circularly polarized light (CPL) emission of the OLED involving R/S-D(16)CzCN as host.*

Response: We thank the reviewer for raising this critical point. The CPL emission in OLEDs employing R/S-D(16)CzCN as the chiral host stems from a synergistic interplay between two key processes: (i) Förster resonance energy transfer (FRET) from the chiral host to the achiral MR-TADF emitter and (ii) diastereomeric interaction (a host-guest interaction that energetically favor specific conformation of the emitter). The chiral host R/S-D(16)CzCN creates a chiral microenvironment around the emitter BN2. During FRET, dynamic structural features in BN2 (e.g., diphenylamine rotors) enhance these interactions, promoting a preferential chiral conformation of BN2 and thus leading to CPL emission form BN2. Conversely, emitters lacking such

rotors (DtBuCzB, BNSeSe) exhibit weaker interaction with the host, resulting in markedly reduced CPL emission despite comparable energy transfer efficiency. These findings, detailed in the Supplementary Note 4, confirm that robust CPL in CP-OLEDs based on R/S-D(16)CzCN:BN2 requires deuteration of the chiral host (enhancing its intrinsic chiroptical properties) and the presence of dynamic structural motifs (e.g., rotors) in the achiral emitter, besides efficient FRET.

Additional section (Supplementary Information, Supplementary Note 4):

“Supplementary Note 4. Mechanisms and rationale underlying CPL emission in CP-OLEDs based on R/S-D(16)CzCN as host.

The CPL emission in OLEDs employing R/S-D(16)CzCN as the chiral host arises from a synergistic interplay between two key processes: (i) Förster resonance energy transfer (FRET) from the chiral host to the achiral MR-TADF emitter, and (ii) diastereomeric interactions, which are host-guest interactions that energetically favor specific conformations of the emitter. The chiral host R/S-D(16)CzCN creates an chiral microenvironment around the emitter BN2. During FRET, dynamic structural features in BN2, such as its diphenylamine rotors, enhance these interactions, promoting a preferential chiral conformation of BN2 and thus leading to CPL emission from BN2. Conversely, emitters lacking such rotors (e.g., DtBuCzB and BNSeSe) exhibit weaker interaction with the host, resulting in markedly reduced CPL emission despite comparable energy transfer efficiency. These findings confirm that robust CPL in CP-OLEDs based on R/S-D(16)CzCN:BN2 requires both the deuteration of the chiral host to enhance its chiroptical properties, as well as the presence of dynamic structural motifs (e.g., rotors) in the achiral emitter, besides efficient FRET.”

Comment 8: *Finally, the author's state: "This work offers a universally applicable strategy for the development of high-performance CPL materials by simple deuteration." In my opinion, to substantiate this claim, the authors should demonstrate that the CPL enhancement is valid for at least one other example of a CPL-active molecule (at least in solution or thin film state).*

Response: We appreciate the reviewer for highlighting the importance of validating the universality of our deuteration strategy. In the updated version of our manuscript, we have designed and synthesized another deuterated chiral emitters, (P,P)/(M,M)-D(32)CzTBCO (Fig. 5), structurally distinct from our original chiral emitters. Solution-phase CPL studies across various solvents (hexane, toluene, dichloromethane, and methanol) demonstrated that deuteration enhances both the maximum and average $|g_{PL}|$ by up to 1.4-fold within the 480-600 nm range compared to the hydrogenated counterpart (Supplementary Fig. 32 and Supplementary Table 5). Importantly, this CPL enhancement was further confirmed in solid-state devices: CP-OLEDs using

(P,P)/(M,M)-D(32)CzTBCO as the chiral emitter achieved a 1.5-fold higher EQE (18.9%) and a 1.3-fold increased average $|g_{EL}|$ value (1.0×10^{-2}) compared to the hydrogenated analogue (Fig. 5, Supplementary Figs. 33 and 34). Collectively, these results, combined with our original deuterated chiral emitters, substantiate our claim that CPL enhancement *via* deuteration is not molecule-specific; rather, it represents a generalizable strategy applicable to diverse CPL-active molecules in both solution and thin-film states.

Additional paragraph, (manuscript, page 14):

“To demonstrate the generality of our deuteration strategy for CPL enhancement, we designed another structurally rigid deuterated CPL molecule, (P,P)/(M,M)-D(32)CzTBCO (Fig. 5 and Supplementary Fig. 29-31). As anticipated, this deuterated CPL material exhibited significantly enhanced $|g_{PL}|$ values across various dilute solutions (Supplementary Fig. 32 and Supplementary Table 5). Furthermore, when employed as chiral emitters in CP-OLEDs, the devices based on (P,P)/(M,M)-D(32)CzTBCO achieved an EQE of 18.9% and an average $|g_{EL}|$ value of 1.0×10^{-2} in the wavelength range between 480 and 600 nm (Fig. 5 and Supplementary Fig. 33). Compared with the hydrogenated counterpart (Supplementary Fig. 34), the EQE and $|g_{EL}|$ values were increased by 1.5-fold and 1.3-fold, respectively.”

Fig. 5 | Another example demonstrating the effectiveness of the deuteration strategy. a, Device structure of CP-OLEDs using (P,P)/(M,M)-CzTBCO and (P,P)/(M,M)-D(32)CzTBCO as the chiral emitters. **b**,

Chemical structures of (P,P)-CzTBCO and (P,P)-D(32)CzTBCO. **c,d**, CP-EL performance of CP-OLEDs based on (P,P)/(M,M)-CzTBCO and (P,P)/(M,M)-D(32)CzTBCO as the chiral emitters: CP-EL spectra (**c**); g_{EL} values versus wavelength curves (**d**).

Supplementary Fig. 32. Photophysical and chiroptical properties of (P,P)/(M,M)-CzTBCO and (P,P)/(M,M)-D(32)CzTBCO in different solvents (1×10^{-5} M) at 300 K. (a) PL spectra of (P,P)/(M,M)-CzTBCO in different solvents (1×10^{-5} M) at 300 K. (b) PL spectra of (P,P)/(M,M)-D(32)CzTBCO in different solvents (1×10^{-5} M) at 300 K. (c) CP-PL spectra of (P,P)-CzTBCO in different solvents (1×10^{-5} M) at 300 K. (d) CP-PL spectra of (P,P)-D(32)CzTBCO in different solvents (1×10^{-5} M) at 300 K. (e) g_{PL} values versus wavelength curves of (P,P)-CzTBCO in different solvents (1×10^{-5} M) at 300 K. (f) g_{PL} values versus wavelength curves of (P,P)-D(32)CzTBCO in different solvents (1×10^{-5} M) at 300 K.

at 300 K. (d) CP-PL spectra of (P,P)/(M,M)-D(32)CzTBCO in different solvents (1×10^{-5} M) at 300 K. (e) g_{PL} values versus wavelength curves of (P,P)/(M,M)-CzTBCO in different solvents (1×10^{-5} M) at 300 K. (f) g_{PL} values versus wavelength curves of (P,P)/(M,M)-D(32)CzTBCO in different solvents (1×10^{-5} M) at 300 K.

Supplementary Table 5. Maximum and average g_{PL} values for (P,P)/(M,M)-CzTBCO, and (P,P)/(M,M)-D(32)CzTBCO across various solvents, measured within the 480-600 nm spectral range.

Molecule	Solvent	Max g_{PL} ^a	Average g_{PL} ^a	Max g_{PL} ^b	Average g_{PL} ^b
(P,P)/(M,M)-CzTBCO	hexane	9.1×10^{-3}	8.0×10^{-3}	-8.9×10^{-3}	-7.8×10^{-3}
	toluene	8.6×10^{-3}	7.5×10^{-3}	-8.6×10^{-3}	-7.5×10^{-3}
	dichloromethane	6.4×10^{-3}	5.3×10^{-3}	-6.4×10^{-3}	-5.4×10^{-3}
	methanol	8.1×10^{-3}	6.8×10^{-3}	-8.7×10^{-3}	-6.9×10^{-3}
(P,P)/(M,M)-D(32)CzTBCO	hexane	1.2×10^{-2}	1.1×10^{-2}	-1.2×10^{-2}	-1.1×10^{-2}
	toluene	1.1×10^{-2}	9.2×10^{-3}	-1.1×10^{-2}	-9.1×10^{-3}
	dichloromethane	8.0×10^{-3}	6.6×10^{-3}	-8.3×10^{-3}	-6.7×10^{-3}
	methanol	9.5×10^{-3}	7.9×10^{-3}	-9.4×10^{-3}	-7.9×10^{-3}

^aMaximum g_{PL} values and average g_{PL} values for the (P,P)-enantiomer. ^bMaximum g_{PL} values and average g_{PL} values for the (M,M)-enantiomer.

Supplementary Fig. 33. CP-OLEDs based on (P,P)/(M,M)-D(32)CzTBCO. (a) Current density and

luminance versus driving voltage characteristics. (b) Normalized electroluminescence spectra. (c) EQE versus luminance characteristics. (d) Current and power efficiency versus luminance characteristics.

Supplementary Fig. 34. CP-OLEDs based on (P,P)/(M,M)-CzTBCO. (a) Current density and luminance versus driving voltage characteristics. (b) Normalized electroluminescence spectra. (c) EQE versus luminance characteristics. (d) Current and power efficiency versus luminance characteristics.

Reviewer #2:

General comment 1: *Chen et al. introduced a novel approach for constructing circularly polarized organic light-emitting diodes (CP-OLEDs) by incorporating hydrogen isotopes into their chiral emitters. Their work combines two previously established methods: deuterated TADF emitters and chiral TADF materials based on twisted backbones, which make the LUMO regions chiral, to achieve high-efficiency CP-OLEDs. While the authors present impressive results regarding efficiency and other performance metrics, I do not believe this manuscript is suitable for publication in Nature Communications, which prioritizes highly original scientific contributions.*

The primary method employed, deuteration of chiral TADF, is essentially a combination of two well-established techniques. Neither the deuteration process nor the dissymmetrization of the TADF structure represents a novel contribution by the authors. While it is true that groundbreaking results can sometimes arise from the combination of existing methods, this manuscript does not seem to fit that paradigm. Deuterated TADF emitters have been explored as early as 2021 (see <https://onlinelibrary.wiley.com/doi/full/10.1002/smsc.202000057>), and the efficiency of these systems has steadily improved, with some reaching external quantum efficiencies (EQE) of over 30% (e.g., <https://onlinelibrary.wiley.com/doi/full/10.1002/adom.202300981> and <https://onlinelibrary.wiley.com/doi/10.1002/adom.202401391>).

Response: Thank you for your critical comment. While deuterated TADF emitters have been reported previously, the enhancement of the luminescence dissymmetry factor (g) through deuteration is a novel finding of our study. More importantly, this finding addresses a fundamental and long-standing limitation in CPL material design: boosting emission quantum yield (Φ) ordinarily requires a larger electric dipole transition moment, which inevitably reduces g . Our research demonstrates that strategic deuteration effectively breaks this inherent trade-off, concurrently enhancing both Φ and g of CPL materials.

To rigorously substantiate this novel finding, we have designed and synthesized an additional structurally distinct deuterated CPL emitter. Devices based on this new CPL emitter not only exhibited a high EQE but also achieved a notably increased g_{EL} value of up to 1.0×10^{-2} , further supporting the generality and robustness of our strategy. Additionally, the updated manuscript includes CPL data across multiple solvents, providing comprehensive experimental validation into the relationship between deuteration and CPL properties. We hope our additional experimental evidence have thoroughly addressed the reviewer's concerns and demonstrated the manuscript's suitability for publication in *Nature Communications*.

General comment 2: *It is unclear whether the authors intend to emphasize the circularly polarized (CP) aspect or the OLED efficiency in this work, as their title suggests the former, but the content does not strongly support either focus. They base their system on the R/S-CzCN compound, which was first reported in Angewandte Chemie (<https://onlinelibrary.wiley.com/doi/full/10.1002/anie.201914249>). It is problematic that this critical reference is only cited in ‘method-related reference [ref 51]’, rather than being discussed in the main body of the text. Moreover, the original publication reports much higher g_{EL} values (-1.2×10^{-2} and $+1.4 \times 10^{-2}$) than those presented in this manuscript, which are 20-40 times lower. This significant discrepancy is not addressed in the current submission.*

While I commend the authors for achieving efficiencies as high as 30% or even approaching 40%, the low g_{EL} values raise concerns if the paper’s goal is to present new chiral emission mechanisms. Comparing their deuterated molecule to the non-deuterated version from the Angewandte work, it seems likely that deuteration has significantly diminished the chiroptical response (from 1.2×10^{-2} to $+6.4 \times 10^{-3}$). The authors might be better served by highlighting the non-chiral performance and optimization strategies for high-efficiency OLEDs, as the current results suggest this work aligns more with “high-efficiency OLEDs with weak circular polarization.”

Response: We thank the reviewer for raising these critical points. In the updated version of our manuscript, we have relocated the citation for R/S-CzCN (formerly ref. 51, now ref. 40) to the main text. The apparent discrepancy in g_{EL} values arises from differences in measurement methodologies: ref. 40 (originally ref. 51) employed a quarter-wave plate/linear polarizer setup, which is prone to optical artifacts that may artificially inflate g_{PL} or g_{EL} values, as corroborated by recent studies (*Nat. Commun.* 14, 1065 (2023); *Adv. Mater.* 35, 2302279 (2023)). In contrast, all g_{EL} values in our study were measured consistently using the JASCO CPL-300 spectrometer, which is the commercially instrument recognized for reliable and reproducible CP-EL measurements. Therefore, a direct comparison between our results and those reported in ref. 40 would not be scientifically appropriate (please refer to our response to **Comment 5** for Reviewer #1).

When measured using the JASCO CPL-300 spectrometer, the R/S-CzCN-based CP-OLEDs exhibited an average $|g_{EL}|$ value of only 3.2×10^{-3} . After strategic deuteration, the corresponding devices showed a twofold improvement, reaching an average $|g_{EL}|$ as high as 6.4×10^{-3} , clearly demonstrating that deuteration promotes enhanced CPL emission. To further substantiate this finding, we synthesized another structurally distinct deuterated CPL emitter, (P,P)/(M,M)-D(32)CzTBCO, achieving a 1.3-fold enhancement of average $|g_{EL}|$ value (1.0×10^{-2}) compared to the hydrogenated analogue. Collectively, these results demonstrate that our deuteration strategy significantly enhances circular polarization in CP-OLEDs.

General comment 3: *In the context of chiral OLEDs and circularly polarized luminescence (CPL), the community is in need of systems that push g_{EL} values to 10^{-2} or higher. The benchmark for g_{EL} has been set at approximately 10^{-3} since 2015, particularly with $EQE > 30\%$ systems that can be achieved by simply separating commercially available emitters (<https://www.nature.com/articles/srep14912>).*

Response: We appreciate the reviewer for the professional comment. In response to the concern raised, we strategically designed and synthesized (P,P)/(M,M)-D(32)CzTBCO (Fig. 5, Supplementary Fig. 32 and Supplementary Table 5), a pair of chiral emitters integrating deuteration to enhance CPL. When employed into CP-OLEDs, these deuterated emitters enabled devices to achieve an EQE of 18.9% and an average $|g_{EL}|$ value of 1.0×10^{-2} in the wavelength range between 480 and 600 nm (Fig. 5 and Supplementary Fig. 33). These results demonstrate that our deuteration strategy is not only generalizable but also effective in pushing g_{EL} values toward 10^{-2} while maintaining high device efficiency.

Additional paragraph, (manuscript, page 14):

“To demonstrate the generality of our deuteration strategy for CPL enhancement, we designed another structurally rigid deuterated CPL molecule, (P,P)/(M,M)-D(32)CzTBCO (Fig. 5 and Supplementary Figs. 29-31). As anticipated, this deuterated CPL material exhibited significantly enhanced $|g_{PL}|$ values across various dilute solutions (Supplementary Fig. 32 and Supplementary Table 5). Furthermore, when employed as chiral emitters in CP-OLEDs, the devices based on (P,P)/(M,M)-D(32)CzTBCO achieved an EQE of 18.9% and an average $|g_{EL}|$ value of 1.0×10^{-2} in the wavelength range between 480 and 600 nm (Supplementary Fig. 33). Compared with the hydrogenated counterpart (Supplementary Fig. 34), the EQE and $|g_{EL}|$ values were increased by 1.5-fold and 1.3-fold, respectively.”

Fig. 5 | Another example demonstrating the effectiveness of the deuteration strategy. a, Device structure of CP-OLEDs using (P,P)/(M,M)-CzTBCO and (P,P)/(M,M)-D(32)CzTBCO as the chiral emitters. **b**, Chemical structures of (P,P)-CzTBCO and (P,P)-D(32)CzTBCO. **c,d**, CP-EL performance of CP-OLEDs based on (P,P)/(M,M)-CzTBCO and (P,P)/(M,M)-D(32)CzTBCO as the chiral emitters: CP-EL spectra (**c**); g_{EL} values versus wavelength curves (**d**).

Supplementary Fig. 32. Photophysical and chiroptical properties of (P,P)/(M,M)-CzTBCO and (P,P)/(M,M)-D(32)CzTBCO in different solvents (1×10^{-5} M) at 300 K. (a) PL spectra of (P,P)/(M,M)-CzTBCO in different solvents (1×10^{-5} M) at 300 K. (b) PL spectra of (P,P)/(M,M)-D(32)CzTBCO in different solvents (1×10^{-5} M) at 300 K. (c) CP-PL spectra of (P,P)/(M,M)-CzTBCO in different solvents (1×10^{-5} M) at 300 K. (d) CP-PL spectra of (P,P)/(M,M)-D(32)CzTBCO in different solvents (1×10^{-5} M) at 300 K. (e) g_{PL} values versus wavelength curves of (P,P)/(M,M)-CzTBCO in different solvents (1×10^{-5} M) at 300 K. (f) g_{PL} values versus wavelength curves of (P,P)/(M,M)-D(32)CzTBCO in different solvents (1×10^{-5} M) at 300 K.

Supplementary Table 5. Maximum and average g_{PL} values for (P,P)/(M,M)-CzTBCO, and (P,P)/(M,M)-D(32)CzTBCO across various solvents, measured within the 480-600 nm spectral range.

Molecule	Solvent	Max g_{PL}^a	Average g_{PL}^a	Max g_{PL}^b	Average g_{PL}^b
(P,P)/(M,M)-CzTBCO	hexane	9.1×10^{-3}	8.0×10^{-3}	-8.9×10^{-3}	-7.8×10^{-3}
	toluene	8.6×10^{-3}	7.5×10^{-3}	-8.6×10^{-3}	-7.5×10^{-3}
	dichloromethane	6.4×10^{-3}	5.3×10^{-3}	-6.4×10^{-3}	-5.4×10^{-3}
	methanol	8.1×10^{-3}	6.8×10^{-3}	-8.7×10^{-3}	-6.9×10^{-3}
(P,P)/(M,M)-D(32)CzTBCO	hexane	1.2×10^{-2}	1.1×10^{-2}	-1.2×10^{-2}	-1.1×10^{-2}
	toluene	1.1×10^{-2}	9.2×10^{-3}	-1.1×10^{-2}	-9.1×10^{-3}
	dichloromethane	8.0×10^{-3}	6.6×10^{-3}	-8.3×10^{-3}	-6.7×10^{-3}
	methanol	9.5×10^{-3}	7.9×10^{-3}	-9.4×10^{-3}	-7.9×10^{-3}

^aMaximum g_{PL} values and average g_{PL} values for the (P,P)-enantiomer. ^bMaximum g_{PL} values and average g_{PL} values for the (M,M)-enantiomer.

Supplementary Fig. 33. CP-OLEDs based on (P,P)/(M,M)-D(32)CzTBCO. (a) Current density and luminance versus driving voltage characteristics. (b) Normalized electroluminescence spectra. (c) EQE versus luminance characteristics. (d) Current and power efficiency versus luminance characteristics.

Supplementary Fig. 34. CP-OLEDs based on (P,P)/(M,M)-CzTBCO. (a) Current density and luminance versus driving voltage characteristics. (b) Normalized electroluminescence spectra. (c) EQE versus luminance characteristics. (d) Current and power efficiency versus luminance characteristics.

Reviewer #3:

General comment 1: *The authors describe the design of chiral emitters displaying circularly polarized thermally activated delayed fluorescence (CP-TADF) and explore how deuteration of the donor unit can potentially impact the photophysical and chiroptical properties of these emitters. They compare the optoelectronic properties of three different systems and conclude that deuteration of the emitters leads to an increase in both the photoluminescence quantum yield (PLQY) and the intensity of circularly polarized luminescence (CPL). They also demonstrate this beneficial effect in CP-OLEDs. While deuteration of organic compounds has already been explored with similar effects on PLQY, the observed increase in CPL intensity (by a factor of 2) is quite novel, in my opinion, and should be of interest to researchers in this field.*

Response: We sincerely appreciate the reviewer's positive comments.

General comment 2: *Although I believe this manuscript could be suitable for publication in Nature Communications, I am curious about how generalizable this strategy is, which I consider a crucial point for Nature research journals. Additionally, I would appreciate seeing more photophysical and chiroptical data recorded in solution to better understand this "deuterium effect."*

Response: We sincerely appreciate the reviewer's constructive feedback. In response, we have designed and synthesized an additional pair of deuterated chiral emitters, (P,P)/(M,M)-D(32)CzTBCO, along with their hydrogenated analogues, (P,P)/(M,M)-CzTBCO, for comparison. Furthermore, we have conducted solution-phase PL and CPL studies for five enantiomer pairs (R/S-MeCzCN, R/S-CzCN, R/S-D(16)CzCN, (P,P)/(M,M)-CzTBCO and (P,P)/(M,M)-D(32)CzTBCO) across four solvents (hexane, toluene, DCM, methanol) at 10^{-5} M concentration. The results, now included in **Supplementary Figs. 11 and 32** as well as **Supplementary Table 3 and 5**, consistently show that the deuterated emitters exhibit increased average $|g_{\text{PL}}|$ values compared to their protonated counterparts. These findings clearly support that deuteration effectively suppresses molecular vibrations and thereby promotes CPL emission, reinforcing the general applicability of our strategy.

Additional sentence, (manuscript, page 8): **“Systematic CPL studies of all three CP-TADF enantiomer pairs in dilute solutions (1×10^{-5} M) at 300 K (Supplementary Fig. 11 and Supplementary Table 3) further confirmed that deuteration enhances the g_{PL} of chiral emitters.”**

Additional paragraph, (manuscript, page 14):

“To demonstrate the generality of our deuteration strategy for CPL enhancement, we designed another structurally rigid deuterated CPL molecule, (P,P)/(M,M)-D(32)CzTBCO (Fig. 5 and Supplementary Figs. 29-

31). As anticipated, this deuterated CPL material exhibited significantly enhanced $|g_{PL}|$ values across various dilute solutions (Supplementary Fig. 32 and Supplementary Table 5). Furthermore, when employed as chiral emitters in CP-OLEDs, the devices based on (P,P)/(M,M)-D(32)CzTBCO achieved an EQE of 18.9% and an average $|g_{EL}|$ value of 1.0×10^{-2} in the wavelength range between 480 and 600 nm (Supplementary Fig. 33). Compared with the hydrogenated counterpart (Supplementary Fig. 34), the EQE and $|g_{EL}|$ values were increased by 1.5-fold and 1.3-fold, respectively.”

Supplementary Fig. 11. Photophysical and chiroptical properties of R/S-MeCzCN, R/S-CzCN, and R/S-D(16)CzCN in different solvents (1×10^{-5} M) at 300 K. (a) PL spectra of R/S-MeCzCN in different solvents (1×10^{-5} M) at 300 K. (b) PL spectra of R/S-CzCN in different solvents (1×10^{-5} M) at 300 K. (c) PL spectra of R/S-D(16)CzCN in different solvents (1×10^{-5} M) at 300 K. (d) CP-PL spectra of R/S-MeCzCN in different solvents (1×10^{-5} M) at 300 K. (e) CP-PL spectra of R/S-CzCN in different solvents (1×10^{-5} M) at 300 K. (f) CP-PL spectra of R/S-D(16)CzCN in different solvents (1×10^{-5} M) at 300 K. (g) g_{PL} values versus wavelength

curves of R/S-MeCzCN in different solvents (1×10^{-5} M) at 300 K. (h) g_{PL} values versus wavelength curves of R/S-CzCN in different solvents (1×10^{-5} M) at 300 K. (i) g_{PL} values versus wavelength curves of R/S-D(16)CzCN in different solvents (1×10^{-5} M) at 300 K.

Supplementary Table 3. Maximum and average g_{PL} values for R/S-MeCzCN, R/S-CzCN, and R/S-D(16)CzCN across various solvents, measured within the 430-580 nm spectral range.

Molecule	Solvent	Max g_{PL} ^a	Average g_{PL} ^a	Max g_{PL} ^b	Average g_{PL} ^b
R/S-MeCzCN	hexane	2.9×10^{-3}	1.5×10^{-3}	-2.8×10^{-3}	-1.8×10^{-3}
	toluene	2.2×10^{-3}	1.7×10^{-3}	-2.5×10^{-3}	-1.9×10^{-3}
	dichloromethane	2.3×10^{-3}	1.7×10^{-3}	-2.9×10^{-3}	-1.9×10^{-3}
	methanol	4.3×10^{-3}	1.8×10^{-3}	-3.6×10^{-3}	-1.7×10^{-3}
R/S-CzCN	hexane	4.2×10^{-3}	3.3×10^{-3}	-4.5×10^{-3}	-3.1×10^{-3}
	toluene	4.3×10^{-3}	3.3×10^{-3}	-4.8×10^{-3}	-3.3×10^{-3}
	dichloromethane	4.1×10^{-3}	3.3×10^{-3}	-4.0×10^{-3}	-3.3×10^{-3}
	methanol	3.9×10^{-3}	2.9×10^{-3}	-4.2×10^{-3}	-3.1×10^{-3}
R/S-D(16)CzCN	hexane	7.5×10^{-3}	6.0×10^{-3}	-8.0×10^{-3}	-5.8×10^{-3}
	toluene	7.1×10^{-3}	6.1×10^{-3}	-7.3×10^{-3}	-5.8×10^{-3}
	dichloromethane	8.5×10^{-3}	5.9×10^{-3}	-8.0×10^{-3}	-6.1×10^{-3}
	methanol	7.1×10^{-3}	5.4×10^{-3}	-1.0×10^{-2}	-5.5×10^{-3}

^aMaximum g_{PL} values and average g_{PL} values for the R-enantiomer. ^bMaximum g_{PL} values and average g_{PL} values for the S-enantiomer.

Supplementary Fig. 32. Photophysical and chiroptical properties of (P,P)/(M,M)-CzTBCO and (P,P)/(M,M)-D(32)CzTBCO in different solvents (1×10^{-5} M) at 300 K. (a) PL spectra of (P,P)/(M,M)-CzTBCO in different solvents (1×10^{-5} M) at 300 K. (b) PL spectra of (P,P)/(M,M)-D(32)CzTBCO in different solvents (1×10^{-5} M) at 300 K. (c) CP-PL spectra of (P,P)/(M,M)-CzTBCO in different solvents (1×10^{-5} M) at 300 K. (d) CP-PL spectra of (P,P)/(M,M)-D(32)CzTBCO in different solvents (1×10^{-5} M) at 300 K. (e) g_{PL} values versus wavelength curves of (P,P)/(M,M)-CzTBCO in different solvents (1×10^{-5} M) at 300 K. (f) g_{PL} values versus wavelength curves of (P,P)/(M,M)-D(32)CzTBCO in different solvents (1×10^{-5} M) at 300 K.

Supplementary Table 5. Maximum and average g_{PL} values for (P,P)/(M,M)-CzTBCO, and (P,P)/(M,M)-D(32)CzTBCO across various solvents, measured within the 480-600 nm spectral range.

Molecule	Solvent	Max g_{PL} ^a	Average g_{PL} ^a	Max g_{PL} ^b	Average g_{PL} ^b
(P,P)/(M,M)-CzTBCO	hexane	9.1×10^{-3}	8.0×10^{-3}	-8.9×10^{-3}	-7.8×10^{-3}
	toluene	8.6×10^{-3}	7.5×10^{-3}	-8.6×10^{-3}	-7.5×10^{-3}
	dichloromethane	6.4×10^{-3}	5.3×10^{-3}	-6.4×10^{-3}	-5.4×10^{-3}
	methanol	8.1×10^{-3}	6.8×10^{-3}	-8.7×10^{-3}	-6.9×10^{-3}
(P,P)/(M,M)-D(32)CzTBCO	hexane	1.2×10^{-2}	1.1×10^{-2}	-1.2×10^{-2}	-1.1×10^{-2}
	toluene	1.1×10^{-2}	9.2×10^{-3}	-1.1×10^{-2}	-9.1×10^{-3}
	dichloromethane	8.0×10^{-3}	6.6×10^{-3}	-8.3×10^{-3}	-6.7×10^{-3}
	methanol	9.5×10^{-3}	7.9×10^{-3}	-9.4×10^{-3}	-7.9×10^{-3}

^aMaximum g_{PL} values and average g_{PL} values for the (P,P)-enantiomer. ^bMaximum g_{PL} values and average g_{PL} values for the (M,M)-enantiomer.

Comment 1: Here are some comments that may be of interest to the authors: Overall, I feel that certain turns of phrase used in the manuscript could be revised for clarity. Moreover, the cited publications on CPL intensity in organic systems are heavily focused on CP-TADF emitters. While I understand this focus, I believe that papers describing CPL emitters with g_{lum} values around $\sim 10^{-2}$ also merit citation. For example: P. Reine et al., Chiral double-stapled o-OPEs with intense circularly polarized luminescence, *Chem. Commun. (Camb)*, 2019, 55, 10685; K. Dhbaibi et al., Achieving high circularly polarized luminescence with push-pull helicenic systems: from rationalized design to top-emission CP-OLED applications, *Chem. Sci.*, 2021, 12, 5522.

Response: According to your suggestion, we have revised several phrases in the manuscript. For example, “emission efficiency” has been corrected to “emission quantum yield,” “electric transition dipole moment” to “electric dipole transition moment,” and “magnetic transition dipole moment” to “magnetic dipole transition moment.” Additionally, we have cited the recommended references in our manuscript.

Additional sentences, (manuscript, page 18): “26. Reine, P. et al. Chiral double stapled o-OPEs with intense circularly polarized luminescence. *Chem. Commun.* **55**, 10685-10688 (2019).

27. Dhbaibi, K. et al. Achieving high circularly polarized luminescence with push-pull helicenic systems: from rationalized design to top-emission CP-OLED applications. *Chem. Sci.* **12**, 5522-5533 (2021).”

Comment 2: Have the authors explored the solvatochromism of these CPL emitters using solvents with different polarities, for both luminescence and CPL measurements?

Response: We appreciate the reviewer’s professional suggestion. In response, we have conducted solution-phase PL and CPL studies for three original CPL emitters (R/S-MeCzCN, R/S-CzCN, and R/S-D(16)CzCN) and two newly synthesized CPL emitters ((P,P)/(M,M)-CzTBCO and (P,P)/(M,M)-D(32)CzTBCO) in four solvents of increasing polarity (hexane, toluene, DCM, methanol) at 10^{-5} M concentration. As shown in

Supplementary Figs. 11 and 32, the PL measurements reveal a clear solvatochromism: increasing solvent polarity leads to a gradual red-shift in the emission maxima and a broadening of the emission bandwidth (quantified by increased full-width-at-half-maximum, FWHM). Crucially, the CPL measurements exhibit the same trend as observed in the solid state: the average $|g_{\text{PL}}|$ values (in the wavelength range of 430-580 nm) increase progressively from $1.8 \pm 0.1 \times 10^{-3}$ for R/S-MeCzCN to $3.2 \pm 0.1 \times 10^{-3}$ for R/S-CzCN, and further to $5.8 \pm 0.2 \times 10^{-3}$ for R/S-D(16)CzCN.

Supplementary Fig. 11. Photophysical and chiroptical properties of R/S-MeCzCN, R/S-CzCN, and R/S-D(16)CzCN in different solvents (1×10^{-5} M) at 300 K. (a) PL spectra of R/S-MeCzCN in different solvents (1×10^{-5} M) at 300 K. (b) PL spectra of R/S-CzCN in different solvents (1×10^{-5} M) at 300 K. (c) PL spectra of R/S-D(16)CzCN in different solvents (1×10^{-5} M) at 300 K. (d) CP-PL spectra of R/S-MeCzCN in different solvents (1×10^{-5} M) at 300 K. (e) CP-PL spectra of R/S-CzCN in different solvents (1×10^{-5} M) at 300 K. (f) CP-PL spectra of R/S-D(16)CzCN in different solvents (1×10^{-5} M) at 300 K. (g) g_{PL} values versus wavelength

curves of R/S-MeCzCN in different solvents (1×10^{-5} M) at 300 K. (h) g_{PL} values versus wavelength curves of R/S-CzCN in different solvents (1×10^{-5} M) at 300 K. (i) g_{PL} values versus wavelength curves of R/S-D(16)CzCN in different solvents (1×10^{-5} M) at 300 K.

Supplementary Fig. 32. Photophysical and chiroptical properties of (P,P)/(M,M)-CzTBCO and (P,P)/(M,M)-D(32)CzTBCO in different solvents (1×10^{-5} M) at 300 K. (a) PL spectra of (P,P)/(M,M)-

CzTBCO in different solvents (1×10^{-5} M) at 300 K. (b) PL spectra of (P,P)/(M,M)-D(32)CzTBCO in different solvents (1×10^{-5} M) at 300 K. (c) CP-PL spectra of (P,P)/(M,M)-CzTBCO in different solvents (1×10^{-5} M) at 300 K. (d) CP-PL spectra of (P,P)/(M,M)-D(32)CzTBCO in different solvents (1×10^{-5} M) at 300 K. (e) g_{PL} values versus wavelength curves of (P,P)/(M,M)-CzTBCO in different solvents (1×10^{-5} M) at 300 K. (f) g_{PL} values versus wavelength curves of (P,P)/(M,M)-D(32)CzTBCO in different solvents (1×10^{-5} M) at 300 K.

Comment 3: *I find the connection between references 9 and 36 and the sentence, "Additionally, it is known that the deuterium nucleus (deuteron), comprising a proton and a neutron, has a ground-state total spin of 1 (ref. 35). Given that the spin contribution to m (which is positively correlated with g)..." unclear. Could the authors provide more details here?*

Response: We thank the reviewer for this valuable suggestion. The deuterium nucleus (deuteron) consists of a proton and a neutron. Due to their nearly identical masses and nuclear properties, the proton and neutron can be regarded as two quantum states of the same particle type (the nucleon). Although only the proton carries electric charge, this difference becomes negligible in nuclear systems because the strong nuclear interaction greatly surpasses the electromagnetic interaction. The strong nuclear attraction in the nucleon pair stabilizes the deuteron, resulting in a nuclear ground-state spin of 1. Additionally, as shown explicitly by Eq. S1 in

Supplementary Note 1, the spin contribution to the magnetic dipole transition moment (\vec{S}_n) is positively correlated with the g -factor, thereby linking nuclear spin states to observed chiroptical properties.

Additional section (Supplementary Information, Supplementary Note 1):

“Supplementary Note 1. Spin contribution to the magnetic dipole transition moment.

Proton and neutron, sharing nearly identical masses and nuclear properties, are described as two quantum states of a single entity (the nucleon). Although only proton carries electric charge, the difference between proton and neutron becomes negligible in nuclear systems due to the dominance of the strong nuclear interaction over electromagnetic interaction. The stronger nuclear attraction in the nucleon pair stabilizes the deuterium nucleus (deuteron), yielding a ground-state nuclear spin of 1.

The g -factor, a dimensionless parameter quantifying the CPL magnitude, can be estimated as follows:

$$g = \frac{4R}{D} = \frac{4 \langle \Psi_{S_1} | \vec{\mu}_e | \Psi_{S_0} \rangle \cdot \langle \Psi_{S_1} | \left(\frac{i}{2c} \right) \vec{m} | \Psi_{S_0} \rangle}{D} = \frac{4 \langle \Psi_{S_1} | -e\vec{r} | \Psi_{S_0} \rangle \cdot \langle \Psi_{S_1} | \left(\frac{-ie}{2m_e \cdot c} \right) \vec{L}_n + g_e \vec{S}_n | \Psi_{S_0} \rangle}{D}. \quad (S1)$$

where e , the elementary charge of an electron; \vec{r} , the position operator; m_e is the mass of the electron; c , the speed of light; \vec{L} , orbital angular momentum; $g_e \approx 2$, the free electron g -factor; \vec{S} , spin angular momentum; n is the components of the Cartesian coordinate system. The term (\vec{S}_n) specifically represents the spin contribution to the magnetic dipole transition moment. From Eq. S1, it is clear that the spin contribution (\vec{S}_n) is positively correlated with the g -factor.”

Comment 4: *Have the authors measured the g_{abs} for the low-energy transition of their compounds? It seems that the sign of the latter is reversed when comparing ECD and CPL (i.e., the R enantiomers give positive CPL but negative ECD). Am I correct? Could the authors clarify this point?*

Response: We thank the reviewer for this insightful comment. Upon re-examining the data, we confirm that the low-energy transitions (375-400 nm) exhibit weak but measurable CD signals. After signal amplification (Fig. 1g), we observe that the CD signal of the R/S-enantiomers has the same sign as the corresponding CPL signal. To quantitatively evaluate the magnitude of CD, the average $|g_{abs}|$ values for these transitions (375-400 nm) of R/S-MeCzCN, R/S-CzCN, and R/S-D(16)CzCN were calculated as 2.2×10^{-4} , 2.4×10^{-4} , and 2.9×10^{-4} , respectively.

Additional sentences (Manuscript, page 8): “We quantified the average absorption dissymmetry factor ($|g_{abs}|$) in the low-energy transition range (375-400 nm), with values of 2.2×10^{-4} for R/S-MeCzCN, 2.4×10^{-4} for R/S-CzCN, and 2.9×10^{-4} for R/S-D(16)CzCN.”

Comment 5: *How were the films prepared?*

Response: We appreciate the reviewer’s question. All films used for optical measurements were deposited by thermal evaporation onto quartz substrates under high vacuum ($\leq 5.0 \times 10^{-5}$ Pa). Prior to deposition, the substrates were sequentially ultrasonicated in acetone and ethanol, dried with a nitrogen stream, and treated for 10 min in a UV-ozone processor (PL16 series, Sen Lights Corporation) to remove residual contaminants.

Additional paragraph, (manuscript, page 23):

“**Fabrication of thin films.** Quartz substrates were sequentially cleaned by ultrasonic treatment in acetone and ethanol, followed by drying under a nitrogen gas flow. The substrates were then treated for 10 min using a UV-ozone surface processor (PL16 series, Sen Lights Corporation). Organic thin films for optical measurements were deposited onto the cleaned quartz substrates *via* thermal evaporation under high vacuum

conditions ($\leq 5.0 \times 10^{-5}$ Pa).”

Comment 6: *The authors mention that the radiative decay becomes faster for deuterated emitters while the non-radiative decay decreases, yet the emitters exhibit the same luminescence lifetime. Why is this the case?*

Response: We thank the reviewer for raising this key point. Our data show that the luminescence lifetimes of the three CP-TADF emitters are close but not the same (Supplementary Fig. 12 and Supplementary Table 2). The radiative decay rate constant ($k_{r,s}$) inherently depends on both the measured lifetime (τ_p and τ_d) and photoluminescence quantum yield (Φ_p and Φ_d) via the relationship $k_{r,s} = \frac{\Phi_p}{\tau_p} + \frac{\Phi_d}{\tau_d}$ (Supplementary Note 2). Deuterated emitters exhibit a larger enhancement in Φ_p and Φ_d , which outweighs the slight variations in τ_p and τ_d , resulting in a net increase in $k_{r,s}$.

Additional sentences (Manuscript, page 6): “The PL decay curves (Supplementary Fig. 10) showed prompt/delayed lifetimes of 23.1 ns/4.7 μ s for R/S-MeCzCN, 24.8 ns/4.2 μ s for R/S-CzCN and 23.5 ns/4.5 μ s for R/S-D(16)CzCN.”

Additional section (Supplementary Information, Supplementary Note 2):

“Supplementary Note 2. Analyses of rate constants.

The respective rate constants of the prompt and delayed fluorescence components (k_p and k_d) occurring in the CP-TADF emitter, can be given by:

$$k_p = \frac{k_{r,s} + k_{nr,s} + k_{isc} + k_{r,t} + k_{nr,t} + k_{risc}}{2} \times \left(1 + \sqrt{1 - \frac{4(k_{r,s} + k_{nr,s} + k_{isc})(k_{r,t} + k_{nr,t} + k_{risc}) - 4k_{isc}k_{risc}}{(k_{r,s} + k_{nr,s} + k_{isc} + k_{r,t} + k_{nr,t} + k_{risc})^2}} \right). \quad (S2)$$

$$k_d = \frac{k_{r,s} + k_{nr,s} + k_{isc} + k_{r,t} + k_{nr,t} + k_{risc}}{2} \times \left(1 - \sqrt{1 - \frac{4(k_{r,s} + k_{nr,s} + k_{isc})(k_{r,t} + k_{nr,t} + k_{risc}) - 4k_{isc}k_{risc}}{(k_{r,s} + k_{nr,s} + k_{isc} + k_{r,t} + k_{nr,t} + k_{risc})^2}} \right). \quad (S3)$$

k_p and k_d can be experimentally determined from prompt and delayed fluorescence decay time constants τ_p , τ_d as follows:

$$k_p = \frac{1}{\tau_p}. \quad (S4)$$

$$k_d = \frac{1}{\tau_d}. \quad (S5)$$

Besides, the emission quantum yields Φ_p and Φ_d for the prompt and delayed fluorescence components have the following relationship with these rate constants:

$$\Phi_p = \frac{k_{r,S}}{k_p} \frac{k_{r,S} + k_{nr,S} + k_{ISC} - k_d}{k_p - k_d}. \quad (S6)$$

$$\Phi_d = \frac{k_{r,S}}{k_d} \frac{k_p - k_{r,S} - k_{nr,S} - k_{ISC}}{k_p - k_d}. \quad (S7)$$

From the equations above, one could obtain the following relationship between rate constants and k_p , k_d , Φ_p and Φ_d experimentally determined from typical PLQY and transient PL characteristics:

$$k_{r,S} = \Phi_p k_p + \Phi_d k_d. \quad (S8)$$

$$k^S = k_{r,S} + k_{nr,S} + k_{ISC} = \frac{\Phi_p k_p^2 + \Phi_d k_d^2}{k_{r,S}}. \quad (S9)$$

$$k^T = k_{r,T} + k_{nr,T} + k_{RISC} = \frac{(\Phi_p + \Phi_d) k_p k_d}{k_{r,S}}. \quad (S10)$$

$$k_{ISC} k_{RISC} = \frac{\Phi_p \Phi_d k_p k_d (k_p - k_d)^2}{k_{r,S}^2}. \quad (S11)$$

”

Comment 7: *The reduction in molecular vibrations in the luminescence spectra is not entirely clear to me. Would it be useful to analyze the full width at half maximum (FWHM)?*

Response: We appreciate the reviewer’s insightful comment. The reduced molecular vibrations indeed correlate with a narrower full width at half maximum (FWHM). Specifically, R/S-D(16)CzCN neat films exhibit a smaller FWHM (Fig. 1b), consistent with suppressed molecular vibrations. This trend is further corroborated in dilute solutions (1×10^{-5} M, 300 K) across solvents of varying polarity (Supplementary Fig. 11), where the PL spectra similarly show a clear reduction in FWHM with decreasing molecular vibrations.

Additional sentence, (manuscript, page 6): “Notably, R/S-D(16)CzCN exhibited a narrower PL emission bandwidth than its non-deuterated analogs (R/S-MeCzCN, R/S-CzCN), a trend further confirmed in dilute solutions (1×10^{-5} M, 300 K; Supplementary Fig. 11).”

Supplementary Fig. 11. Photophysical and chiroptical properties of R/S-MeCzCN, R/S-CzCN, and R/S-D(16)CzCN in different solvents (1×10^{-5} M) at 300 K. (a) PL spectra of R/S-MeCzCN in different solvents (1×10^{-5} M) at 300 K. (b) PL spectra of R/S-CzCN in different solvents (1×10^{-5} M) at 300 K. (c) PL spectra of R/S-D(16)CzCN in different solvents (1×10^{-5} M) at 300 K. (d) CP-PL spectra of R/S-MeCzCN in different solvents (1×10^{-5} M) at 300 K. (e) CP-PL spectra of R/S-CzCN in different solvents (1×10^{-5} M) at 300 K. (f) CP-PL spectra of R/S-D(16)CzCN in different solvents (1×10^{-5} M) at 300 K. (g) g_{PL} values versus wavelength curves of R/S-MeCzCN in different solvents (1×10^{-5} M) at 300 K. (h) g_{PL} values versus wavelength curves of R/S-CzCN in different solvents (1×10^{-5} M) at 300 K. (i) g_{PL} values versus wavelength curves of R/S-D(16)CzCN in different solvents (1×10^{-5} M) at 300 K.

Comment 8: *The Huang-Rhys analysis is quite interesting. However, CzCn and MeCzCn display similar patterns despite differences in PLQY. How can these results be interpreted?*

Response: We thank the reviewer for this insightful question. While the overall Huang-Rhys patterns of R/S-MeCzCN and R/S-CzCN appear broadly similar, critical differences emerge in the $250\text{-}750\text{ cm}^{-1}$ region

(Figs. 1d,e). R/S-MeCzCN exhibits two prominent vibrational modes with Huang-Rhys factor (S_{ev}) exceeding 0.2: a CH₃-dominated wagging vibration mode at 274 cm⁻¹ ($S_{ev} = 0.33$) and a biphenyl-dominated wagging vibration mode at 471 cm⁻¹ ($S_{ev} = 0.21$). In contrast, R/S-CzCN shows no vibrational modes with Huang-Rhys factors above 0.2 in the same frequency range. These larger Huang-Rhys factors in R/S-MeCzCN correspond to greater normal mode coordinate displacements (Supplementary Fig. 13), which facilitate its non-radiative transition rate ($k_{nr,S}$) and contribute to its lower PLQY compared to R/S-CzCN, as supported by previous literature (*J. Am. Chem. Soc.* 129, 9333-9339 (2007)).

Supplementary Fig. 13. Vector displacement diagram of the normal mode with Huang-Rhys factor exceeding 0.2. (a) Frequency 274 cm⁻¹ plotted on the optimized S₁ geometry of R-MeCzCN. (b) Frequency 471 cm⁻¹ plotted on the optimized S₁ geometry of R-MeCzCN.

Comment 9: *The authors mention the CISS effect in the conclusion, but no experiments related to this phenomenon are presented. What exactly are the authors suggesting here? This is not clear to me.*

Response: We appreciate the reviewer's inquiry and apologize for the abrupt mention to the CISS effect in our conclusion section. We aim to suggest that our proposed deuteration strategy might open new opportunities for spin electronic devices, where deuterated organic chiral materials could potentially enhance spin-related phenomena such as the CISS effect. We have modified this point accordingly in the revised manuscript to avoid any confusion.

Modified sentence, (manuscript, page 16):

“...The proposed design principles **may** also open up new possibilities for **spin electronic devices, where deuterated organic chiral materials could potentially enhance spin-related phenomena such as** chiral-induced spin selectivity...”

Dear Editor and Reviewers,

Enclosed please find our revised manuscript entitled “**Deuteration promotes circularly polarized light emission**” (Manuscript ID: **NCOMMS-24-50744B**) for publication as a research article in *Nature Communications*. We sincerely thank the reviewers for their insightful and professional comments, which helped us substantially improve the quality of our manuscript.

We have meticulously revised the manuscript in response to the reviewers’ comments and the point-by-point responses are provided. In particular: (i) Regarding Reviewer #2’s doubts about the chiral FRET mechanism, we recognize that our initial discussion of diastereomeric interactions was primarily included in the Supplementary Notes, potentially leading to a misunderstanding of our explanation for the chiral host and achiral guest system. To clarify this point, we have now emphasized the role of diastereomeric interactions more explicitly in the manuscript and have included additional experimental and theoretical evidence to support our conclusions. (ii) In response to Reviewer #2’s concern that “key arguments rely heavily on external references”, we have incorporated direct theoretical validation demonstrating the contribution of deuteration to the magnetic transition dipole moment.

For your reference, annotated files of the revised manuscript and supporting information with all changes highlighted in yellow are also attached. We hope that the revised paper could now be satisfied for your requirements.

Yours Sincerely,

Chuluo Yang on behalf of the authors.

Corresponding author: **Professor Chuluo Yang**

at College of Materials Sciences and Engineering, Shenzhen University, Shenzhen 518071, People’s Republic of China. Tel: +86-0755-86713985, e-mail: clyang@szu.edu.cn.

We greatly appreciate the editor and reviewers for taking the time to review and critique our manuscript. Accordingly, we have fully addressed the concerns raised by the reviewers.

Reviewer #1:

General comment: *The authors have significantly improved their manuscript in light of the reviewers' comments. While the pronounced effect of deuteration on the dissymmetry factors observed for CzCN (a twofold enhancement) remains surprising to me, the additional data provided—as well as the demonstration that the g_{lum} enhancement effect is also observable with another pair of CP-TADF emitters (1.3 fold)—are convincing.*

Before I can recommend the publication of this valuable contribution in Nature Communications, I have some final requests.

Response: We sincerely appreciate the reviewer's positive and professional comments.

Comment 1: *On page 8, lines 119–120, the authors mention that they measured g_{abs} values in the range of the S_0 - S_1 transitions approximately ten times lower than g_{lum} values. This seems counterintuitive, unless there is a significant geometrical difference between the ground and excited states. Could the authors clarify or provide a rationale for this observation?*

Response: We thank the reviewer for raising this important point. Upon re-examining the experimental data, we identified an error in the original g_{abs} values directly from the J-1500 CD spectrometer (JASCO). After recalculating g_{abs} using the equation (3):

$$g_{abs} = \frac{2(A_{left} - A_{right})}{A_{left} + A_{right}} \quad (3)$$

where A_{left} and A_{right} are the absorption intensities of the left- and right-circularly polarized light, respectively. The corrected average $|g_{abs}|$ in the wavelength range of 375-400 nm for R/S-MeCzCN, R/S-CzCN, and R/S-D(16)CzCN are 1.2×10^{-3} , 1.5×10^{-3} and 4.1×10^{-3} , respectively. While these g_{abs} values are of the same order of magnitude as the g_{lum} values, they remain smaller. To clarify this discrepancy, we performed CD spectral calculations at the CAM-B3LYP/def2-SVP level based on optimized S_0 geometries. These calculations revealed that the $S_0 \rightarrow S_1$ and $S_0 \rightarrow S_2$ transitions exhibit nearly degenerate energy levels (3.92 eV and 3.93 eV, respectively) but opposite CD signal directions (rotatory strengths: $+40.04 \times 10^{-40}$ esu²·cm² for the $S_0 \rightarrow S_1$ transition, -6.45×10^{-40} esu²·cm² for the $S_0 \rightarrow S_2$ transition; **Supplementary Fig. 15**). Despite the higher rotatory strength of the $S_0 \rightarrow S_1$ transition, the opposite CD signals partially cancel each other, leading to a reduced net CD signal and thus smaller g_{abs} compared to g_{lum} .

Modified sentence, (manuscript, page 8): "... the average absorption dissymmetry factor ($|g_{abs}|$) in the low-energy transition range (375-400 nm), with values of 1.2×10^{-3} for R/S-MeCzCN, 1.5×10^{-3} for R/S-

CzCN, and 4.1×10^{-3} for R/S-D(16)CzCN:

$$g_{abs} = \frac{2(A_{left} - A_{right})}{A_{left} + A_{right}} \quad (3)$$

where A_{left} and A_{right} are the absorption intensities of the left- and right-CPL, respectively (See Supplementary Fig. 15 for simulated CD spectra).”

Supplementary Fig. 15. Simulated CD spectra at the CAM-B3LYP/def2-SVP level based on optimized S_0 geometries. (a) R/S-D(16)CzCN. (b) R/S-CzCN. (c) R/S-MeCzCN.

Comment 2: Regarding the CPL data particularly when comparing the hydrogenated and deuterated CP-TADF emitters, providing the raw data ($\Delta I = f(\lambda)$ curves and the $DC = f(\lambda)$) would be helpful.

Response: We thank the reviewer for this constructive suggestion. As requested, we have included the raw data ($\Delta I = f(\lambda)$ and $DC = f(\lambda)$ curves) for comparison of hydrogenated and deuterated CP-TADF emitters in Supplementary Figs. 17 and 42. Additionally, these raw data are also provided separately as a Source Data file.

Supplementary Fig. 17. ΔI and DC values versus wavelength curves of R/S-CzCN and R/S-D(16)CzCN.

Supplementary Fig. 42. ΔI and DC values versus wavelength curves of (P,P)/(M,M)-CzTBCO and (P,P)/(M,M)-D(32)CzTBCO.

Reviewer #2:

General comment 1: *The authors have made a substantial effort in revising the manuscript by including a significant number of new experimental results, such as additional solution-state CPL measurements and another example of a chiral emitter. I appreciate their responsiveness to the reviewers' concerns and the work invested in this revision. However, despite these additions, I regret to say that I still cannot recommend this manuscript for publication in Nature Communications.*

While the manuscript now includes more data, the core scientific claims remain insufficiently supported. Many of the conclusions are based on speculative reasoning without direct experimental or theoretical validation. In several instances, key arguments rely heavily on external references, some of which contain controversial or unverified claims within the chiral optoelectronics community. This reliance undermines the manuscript's overall scientific robustness.

That said, it is encouraging to see that all three reviewers have highlighted the previously hidden citation of Ref. 51 in the Methods section. It is appropriate that the authors have now integrated this important reference into the main text to properly acknowledge foundational contributions from the community.

Response: We appreciate the reviewer for their critical comment and the opportunity to strengthen our manuscript. To address concerns about the robustness of our core scientific claims, we have conducted additional experimental and theoretical studies that directly support our conclusions, thereby minimizing reliance on external references.

Experimentally, we have measured the CD spectra of R/S-D(16)CzCN as the chiral host combined with various achiral emitters (BN2, DtBuCzB, and BNSeSe), and compared the results with theoretically simulated CD spectra. These data directly confirms that strong diastereomeric interactions are a prerequisite for achieving efficient CP emission from achiral guests in chiral host and achiral guest systems. Additionally, we have also recorded CPL spectra of R/S-D(16)CzCN:BN2 films under different excitation wavelengths (290-410 nm), which provides clear evidence that FRET is the second key condition enabling efficient CP emission in these systems.

On the theoretical front, we have compared the nuclear displacements of the biphenyl skeleton in deuterated and hydrogenated systems and calculated atomic and fragment contributions to the magnetic transition dipole moment (m), as well as its dependence on vibrational frequency. Our results demonstrate that deuteration enhances m primarily by altering the nuclear displacements of the biphenyl backbone, rather than through direct electronic contribution from deuterium atoms to the frontier molecular orbitals.

We believe these combined experimental and theoretical validations fully address the reviewer's concerns,

reinforcing the scientific robustness and suitability of our manuscript for publication in *Nature Communications*.

General comment 2: *Specific comments: The authors report an increase in g_{PL} from 2.9×10^{-3} to 5.8×10^{-3} via deuteration, attributing the enhancement to increased magnetic transition dipole moments (m). However, from Supplementary Fig. S17, there does not appear to be a significant contribution in density distributions from the deuterium atoms. It would strengthen the manuscript if the authors could clarify whether and how these atoms significantly affect the frontier molecular orbitals. If their contribution is indeed minor, it is not possible to attribute such m changes to the deuteration directly and requires better justification.*

Response: We appreciate the reviewer's insightful comment. Our analysis confirms that the transition density is not significantly distributed on the deuterium atoms. Quantitative fuzzy atomic space analysis indicates that, for R/S-D(16)CzCN, the transition density distributions on deuterium atoms is only 2.38%, compared to 1.04% on hydrogen atoms. This demonstrates that the observed increase in the magnetic transition dipole moment (m) upon deuteration does not result directly from the electronic contribution of deuterium atoms to the frontier molecular orbitals.

Instead, the main role of the deuteration is to indirectly alter the nuclear displacement of the biphenyl skeleton (Supplementary Fig. 18). Specifically, geometric optimization considering the Diagonal Born-Oppenheimer correction shows that deuteration shortens bond lengths within the biphenyl backbone, reduces the internal twist of the phenyl rings, and slightly increases the dihedral angle between the two phenyl rings from 46.9° to 47.5°. Since the transition density is predominantly localized on the biphenyl backbone (65.77%), deuteration impacts the magnetic transition dipole moment (m) by altering the nuclear displacement of the biphenyl skeleton. To further substantiate this, we calculated atomic and fragment contributions to m as well as plotted m against vibrational frequency (Supplementary Figs. 19 and 24). Although deuterium atoms slightly increase their direct contribution to m (from 0.02 to 0.03 Bohr magneton), this contribution remains minor, accounting for only 3.6% of the total m (Supplementary Fig. 19). Notably, the biphenyl backbone exhibits a significant enhancement in m (from 0.31 to 0.48 Bohr magneton, a ~1.5-fold increase) and contributes 55.7% to the total m upon deuteration, resulting in a larger m at the stationary point for R/S-D(16)CzCN compared to its hydrogenated analogue. Additionally, when molecular vibrations are considered, the suppression of skeletal vibrations due to deuteration enhances the m associated with vibrational modes in R/S-D(16)CzCN (Supplementary Fig. 24). These findings confirm that deuteration contributes to the observed increase in m predominantly by altering the nuclear displacements of the biphenyl backbone.

We have clarified and emphasized this point in the revised manuscript and supplementary information.

Modified paragraph, (manuscript, page 8): “Additionally, it is known that the deuterium nucleus (deuteron³⁷), comprising a proton and a neutron, **alters molecular skeletal vibrational modes**³⁵. Given that **changes in nuclear displacements can influence the orbital** contribution to m (which is positively correlated with g , Supplementary Note 1)^{9,38}, deuteration may have the potential to enhance g in CPL materials...”

Modified sentence, (manuscript, page 8): “Our results demonstrated that the deuterated molecule R-D(16)CzCN exhibits larger magnetic dipole transition moment values (m) **due to altered nuclear displacements of the biphenyl backbone upon deuteration (Supplementary Figs. 18 and 19)**...”

Modified section (Supplementary Information, Supplementary Note 1):

“**Supplementary Note 1. Orbital contribution to the magnetic transition dipole moment.**

The g -factor, a dimensionless parameter quantifying the CPL magnitude, can be estimated as follows:

$$g = \frac{4R}{D} = \frac{4 \langle \Psi_{S_1} | \vec{\mu}_e | \Psi_{S_0} \rangle \cdot \langle \Psi_{S_1} | \left(\frac{i}{2c} \right) \vec{m}_e | \Psi_{S_0} \rangle}{D} = \frac{4 \langle \Psi_{S_1} | -e\vec{r} | \Psi_{S_0} \rangle \cdot \langle \Psi_{S_1} | \left(\frac{-ie}{2m_e \cdot c} \right) \vec{L}_n + g_e \vec{S}_n | \Psi_{S_0} \rangle}{D} \quad (S1)$$

where e , the elementary charge of an electron; \vec{r} , the position operator; m_e is the mass of the electron; c , the speed of light; \vec{L} , orbital angular momentum; $g_e \approx 2$, the free electron g -factor; \vec{S} , spin angular momentum; n is the components of the Cartesian coordinate system. The term (\vec{S}_n) specifically represents the spin contribution to the magnetic dipole transition moment. From Eq. S1, it is clear that the **orbital contribution** (\vec{L}_n) is positively correlated with the g -factor. **This parameter (\vec{L}_n) is intimately linked to rotating and overlapping between the transition orbitals, which can be modulated by molecular skeletal nuclear displacements.**”

Supplementary Fig. 18. Geometric structural changes of R-CzCN and R-D(16)CzCN. (a) Intuitive numbering sequence for the biphenyl skeleton. **(b)** C-C bond lengths in the biphenyl skeleton. **(c)** Dihedral angle between the two phenyl rings in the biphenyl skeleton. **(d)** Deviation to planar values of the two phenyl rings in the biphenyl skeleton.

Supplementary Fig. 19. Atomic and fragment contributions to m . (a) R-CzCN. (b) R-D(16)CzCN. Note: The scalar sum of fragment $|m|$ values is larger than the whole-molecule m obtained by vector summation, because the fragment m vectors are not collinear and partially cancel.

Supplementary Fig. 24. Theoretically calculated m versus vibrational frequency. (a) R-CzCN. (b) R-D(16)CzCN.

General comment 3: The discussion around so-called “chiral FRET” lacks both theoretical grounding and experimental support. The authors cite a 2013 *Angewandte Chemie* paper as the primary justification, yet that study itself does not offer direct evidence for a chiral FRET mechanism—only a coexistence of CPL and FRET processes. Without detailed calculations or mechanistic insights, it is difficult to accept the implication that a non-chiral emitter in a chiral host can emit CPL at the reported levels. I would consider it is just a result of host material self-absorption or selective scattering, not necessary to be bound with other mechanisms. Known

literature suggests that CPL emission typically requires chirality in the emitter or structural changes in the excited state—conditions not addressed here or in those systems that time-reversal symmetry is broken. David L Andrews' classic paper (<https://iopscience.iop.org/article/10.1088/2050-6120/ab10f0>) clearly demonstrates that achiral emitters cannot generate CPL solely through energy transfer in a chiral host, unless additional symmetry-breaking mechanisms are involved.

If the authors were indeed able to induce CPL in an achiral emitter solely via chiral FRET without invoking such mechanisms, this would constitute a remarkable discovery with implications beyond CPL enhancement, potentially touching on fundamental aspects of quantum electrodynamics or CPT theorem. However, the manuscript does not currently present sufficient evidence to support such a claim.

Conclusion:

Although the manuscript presents some decent OLED performance data (without any doubts), its core contributions to the understanding of molecular chirality and CPL mechanisms remain speculative and, in some cases, potentially misleading. The logical framework often relies on secondary sources without critical evaluation, and the foundational claims require more rigorous theoretical and experimental substantiation. I therefore do not support the publication of this manuscript in its current form.

Response: We thank the reviewer for their critical feedback and regret any ambiguity in our original manuscript. We clarify that our work neither cites the 2013 *Angewandte Chemie* paper nor proposes “chiral FRET” as a standalone mechanism. Instead, we emphasize that, in addition to FRET, strong diastereomeric interactions are the prerequisite for inducing CPL of the achiral BN2 emitter. This is supported by comparative studies involving the same chiral host and different achiral emitters exhibiting distinct diastereomeric interactions, as detailed in the Supplementary Information (**Supplementary Figs. 26-33** and **Supplementary Note 3**). This mechanistic framework is consistent with the symmetry-breaking principles outlined by Andrews *et al.* (*Methods Appl. Fluoresc.* **7**, 032001 (2019)).

In the R/S-D(16)CzCN:BN2 system, BN2 is a racemate composed of two enantiomers (CC-BN2 and C-BN2) that exhibit both propeller chirality (twisted axial blades at $\pm 36\text{--}48^\circ$) and helical chirality arising from its multiresonant framework (**Supplementary Fig. 26**). Density functional theory (DFT) calculations reveal an energetic difference of approximately $4.3 \text{ kcal}\cdot\text{mol}^{-1}$ between the R-D(16)CzCN:CC-BN2 (matched) and R-D(16)CzCN:C-BN2 (mismatched), favoring the former (CC-BN2 enantiomer) under Boltzmann distribution at 298 K ($kT \approx 0.6 \text{ kcal}\cdot\text{mol}^{-1}$). This theoretical prediction is further supported by experimental CD spectra (**Supplementary Fig. 27**). Co-deposited R/S-D(16)CzCN:BN2 films (2 wt% BN2) prepared by vacuum deposition exhibit distinct CD signals in the 400-500 nm range that is absent in the pristine R/S-D(16)CzCN

neat films. Comparison with simulated CD spectra of CC-BN2 and C-BN2 confirms that this signal originates from the induced chiral character of BN2, arising from the ground-state interactions with the neighboring chiral R/S-D(16)CzCN host. To further probe the role of FRET, we measured the CPL spectra of R-D(16)CzCN:BN2 films under excitation at various wavelengths (290-410 nm) and at 500 nm for direct excitation of BN2. Notably, excitation at 290 nm yielded higher g_{PL} values compared to longer wavelengths (310, 330, 350, 370, 390, 410 and 500 nm), confirming the occurrence and contribution of FRET. These results demonstrate that strong diastereomeric interactions, in conjunction with FRET, are both essential for enabling CPL emission from the achiral BN2 emitter within the chiral R/S-D(16)CzCN host.

To underscore the critical role of strong diastereomeric interactions, we also investigated the R/S-D(16)CzCN:DtBuCzB and R/S-D(16)CzCN:BNSeSe systems in the original manuscript. Both DtBuCzB and BNSeSe feature multiresonant frameworks with distorted [4]helicene substructures, with BNSeSe exhibiting greater distortion due to selenium incorporation (Supplementary Figs. 28 and 29). DFT calculations reveal that the energetic difference between the R-D(16)CzCN:*P*-DtBuCzB (matched) and R-D(16)CzCN:*M*-DtBuCzB (mismatched) is minimal, approximately 0.2 kcal·mol⁻¹ (Supplementary Fig. 28). Similarly, the energetic difference between the R-D(16)CzCN:*P*-BNSeSe (matched) and R-D(16)CzCN:*M*-BNSeSe (mismatched) is *ca.* 0.7 kcal·mol⁻¹ (Supplementary Fig. 29). These small energetic differences are insufficient to drive diastereomeric interactions at room temperature, assuming Boltzmann distribution at 298 K ($kT \approx 0.6$ kcal·mol⁻¹). Accordingly, co-deposited R/S-D(16)CzCN:DtBuCzB (2 wt%) and R/S-D(16)CzCN:BNSeSe (2 wt%) films exhibited CD signals indistinguishable from pristine R/S-D(16)CzCN films, with no evidence of induced chirality in the guests. Despite FRET occurring in these systems (Supplementary Fig. 30), their CPL signals remained weak (Supplementary Fig. 33). These results starkly contrast with the R/S-D(16)CzCN:BN2 system, highlighting that strong diastereomeric interactions are essential for achieving efficient CPL in chiral host and achiral guest systems.

We have provided proper discussion in the revised manuscript and supplementary information, and the mentioned reference has been properly cited as Ref. 48.

Modified paragraph, (manuscript, page 12): "...an achiral MR-TADF emitter with strong diastereomeric interaction^{46,47} (for details regarding the selection of achiral guests, see Supplementary Figs. 26-33 and Supplementary Note 3), as the guest emitter. Co-deposited R/S-D(16)CzCN:BN2 films displayed an additional CD signal in the 400-500 nm range, which differed from that of pristine R/S-D(16)CzCN neat films and aligned well with the simulated CD spectra (Supplementary Fig. 27) indicating induced chirality in BN2⁴⁸. Furthermore, the absorption spectrum of BN2 substantially overlapped with the emission spectrum of the R/S-

D(16)CzCN (Fig. 4a), and the CPL spectra of R/S-D(16)CzCN:BN2 films varied systematically with the excitation wavelength (290-410 nm, Supplementary Fig. 27)...”

Modified section (Supplementary Information, Supplementary Note 3):

“Supplementary Note 3. Critical role of diastereomeric interactions.

Strong diastereomeric interactions are prerequisite for achieving efficient CP emission from achiral guest in chiral host and achiral guest systems. Three achiral MR-TADF emitters (BN2, DtBuCzB, and BNSeSe) were selected based on spectral overlap between their absorption and the host’s emission. BN2 exists as a racemate with two enantiomers (*CC*-BN2 and *C*-BN2) exhibiting propeller chirality (twisted axial blades at ± 36 - 48°) and helical chirality derived from its multiresonant framework (Supplementary Fig. 26). In contrast, DtBuCzB and BNSeSe feature distorted [4]helicene substructures, with BNSeSe showing greater distortion due to selenium incorporation (Supplementary Figs. 28 and 29).

Density functional theory (DFT) calculations (Supplementary Fig. 26) reveal an energetic difference of approximately $4.3 \text{ kcal}\cdot\text{mol}^{-1}$ between the R-D(16)CzCN:*CC*-BN2 (matched) and R-D(16)CzCN:*C*-BN2 (mismatched) complexes, thermodynamically favoring the former (*CC*-BN2 enantiomer) under Boltzmann distribution at 298 K ($kT \approx 0.6 \text{ kcal}\cdot\text{mol}^{-1}$). This theoretical prediction is corroborated by experimental CD spectra (Supplementary Fig. 27). R/S-D(16)CzCN:BN2 films (2 wt%) exhibit distinct CD signals in the 400-500 nm range that is absent in the pristine R/S-D(16)CzCN neat films. Comparison with simulated CD spectra of *CC*-BN2 and *C*-BN2 confirms that this signal originates from the induced chiral character of BN2, arising from the ground-state interactions with the neighboring chiral R/S-D(16)CzCN host. In combination with FRET, the R/S-D(16)CzCN:BN2 system enables efficient CPL from the achiral BN2 emitter.

By contrast, DFT calculations indicate that the energetic difference between the R-D(16)CzCN:*P*-DtBuCzB (matched) and R-D(16)CzCN:*M*-DtBuCzB (mismatched) is minimal, approximately $0.2 \text{ kcal}\cdot\text{mol}^{-1}$ (Supplementary Fig. 28). Similarly, the energetic difference between the R-D(16)CzCN:*P*-BNSeSe (matched) and R-D(16)CzCN:*M*-BNSeSe (mismatched) is *ca.* $0.7 \text{ kcal}\cdot\text{mol}^{-1}$ (Supplementary Fig. 29). These small energetic differences are insufficient to drive diastereomeric interactions at room temperature, assuming Boltzmann distribution at 298 K ($kT \approx 0.6 \text{ kcal}\cdot\text{mol}^{-1}$). Accordingly, R/S-D(16)CzCN:DtBuCzB (2 wt%) and R/S-D(16)CzCN:BNSeSe (2 wt%) films display CD signals indistinguishable from pristine R/S-D(16)CzCN films, with no evidence of induced chirality in the guests. Despite the presence of FRET in these systems (Supplementary Fig. 30), their CPL signals remain weak (Supplementary Fig. 33), highlighting the necessity of strong diastereomeric interactions for efficient CPL generation in chiral host and achiral guest systems.”

Additional reference: Ref. 48. Andrews, D. L. Chirality in fluorescence and energy transfer. *Methods Appl. Fluoresc.* 7, 032001 (2019).

Supplementary Fig. 26. BN2 enantiomers and diastereomeric interactions with the chiral host. (a) The chemical structures of BN2 and the optimized geometries of BN2 enantiomers, CC-BN2 and C-BN2. (b) Calculated diastereomeric interactions between R-D(16)CzCN and the BN2 enantiomers (CC-BN2 and C-BN2). Geometries and relative energies were computed at the B3LYP-D3(BJ)/def2-SVP level of theory.

Supplementary Fig. 27. Chiroptical properties of R/S-D(16)CzCN:BN2. (a) Experimental CD spectra of R/S-D(16)CzCN neat films and R/S-D(16)CzCN:BN2 films. (b) Calculated CD spectra of BN2 enantiomers. (c) CP-PL spectra of R/S-D(16)CzCN:BN2 films under excitation at 290, 310, 330, 350, 370, 390 and 410 nm. (d) Column diagrams of $|g_{PL}|$ values of D(16)CzCN:BN2 films excited at 290, 310, 330, 350, 370, 390, 410 and 500 nm.

Supplementary Fig. 28. Diastereomeric interactions and chiroptical properties of R/S-D(16)CzCN:DtBuCzB. (a) Calculated diastereomeric interactions between R-D(16)CzCN and the DtBuCzB enantiomers (*P*-DtBuCzB and *M*-DtBuCzB). Geometries and relative energies were computed at the B3LYP-D3(BJ)/def2-SVP level of theory. (b) Experimental CD spectra of R/S-D(16)CzCN neat films and R/S-D(16)CzCN:DtBuCzB films. (c) Calculated CD spectra of DtBuCzB enantiomers.

Supplementary Fig. 29. Diastereomeric interactions and chiroptical properties of R/S-D(16)CzCN:BNSeSe. (a) Calculated diastereomeric interactions between R-D(16)CzCN and the BNSeSe enantiomers (*P*-BNSeSe and *M*-BNSeSe). Geometries and relative energies were computed at the B3LYP-D3(BJ)/def2-SVP level of theory. (b) Experimental CD spectra of R/S-D(16)CzCN neat films and R/S-D(16)CzCN:BNSeSe films. (c) Calculated CD spectra of BNSeSe enantiomers.

Reviewer #3:

General comment: *Dear authors, Thanks for having addressed most of the concerns raised by the referees. I would be happy to recommend your work for publication.*

A minor concern for me relies on the exact relation between the deuteration and the increase of g_{lum} . While I trust your results, I am just wondering why other chiral deuterated compounds have not shown this effect. It would be nice to provide a comparison to me.

Response: We sincerely appreciate the reviewer's positive feedback and the opportunity to clarify the relationship between deuteration and the increase of g_{lum} (g_{PL}). To our knowledge, few deuterated chiral emitters have been reported. Moreover, no work addressed the issue on relation between the deuteration and the g_{PL} . To explain why some organic chiral emitters show large increases in g_{PL} upon deuteration, while others show modest or negligible changes, we performed comparative calculations on representative chiral emitters (planar chirality: R_p -CzpPhTrz; point chirality: (-)-(R,R)-CAI-Cz; axial chirality: (R)-OBN-Cz; helical chirality: (P)-Aza[6]helicene; **Supplementary Fig. 44**) and the two chiral compounds in our manuscript (R-CzCN and (P,P)-CzTBCO). Theoretical calculations were conducted both at stationary-point TD-DFT (excluding vibrational effects) and with the nuclear ensemble approach (including vibrational effects).

Our results show that, when vibrational effects are included, the average calculated g factor (g_{cal}) from the nuclear ensemble approach decreases relative to the values obtained from stationary-point TD-DFT calculations (which exclude vibrational effects), with g_{cal} reductions ranging from 4.7% to 71.7% (**Supplementary Table 6**). In this work, R/S-CzCN exhibits the largest g_{cal} reduction (71.7%), from 1.2×10^{-2} (without vibrational effects, stationary-point TD-DFT) to 3.4×10^{-3} (with vibrational effects, nuclear ensemble approach). Experimentally, upon deuteration, R/S-D(16)CzCN ($g_{PL} = 5.8 \pm 0.1 \times 10^{-3}$) exhibits a twofold enhancement in g_{PL} compared to R/S-CzCN ($g_{PL} = 2.9 \pm 0.1 \times 10^{-3}$). By contrast, (P,P)/(M,M)-CzTBCO shows a moderate g_{cal} reduction (33.3%), from 1.5×10^{-2} (without vibrational effects, stationary-point TD-DFT) to 1.0×10^{-2} (with vibrational effects, nuclear ensemble approach). Upon deuteration, the experimentally measured g_{PL} enhancement is correspondingly smaller: (P,P)/(M,M)-D(32)CzTBCO ($g_{PL} = 1.1 \times 10^{-2}$) shows a 1.3-fold increase relative to its hydrogenated analogue ($g_{PL} = 8.6 \times 10^{-3}$). Together, these findings suggest that deuteration increases the experimentally observed g_{PL} by mitigating the g_{cal} reductions arising from vibrational effects. In other words, the greater the vibrational effects, the larger the g_{PL} enhancement upon deuteration; conversely, when g_{cal} is minimally affected by molecular vibrations, deuteration may little affect on g_{PL} .

We have incorporated this conclusion into the manuscript.

Modified conclusion, (manuscript, page 16): "...Our findings suggest that deuteration is a strategy to minimize the influence of vibrational modes that reduce the CPL signal. For chiral molecules that are minimally affected by molecular vibrations, deuteration may little affect on g_{PL} (Supplementary Fig. 44 and Supplementary Table 6)..."

Supplementary Fig. 44. Chemical structures of representative organic chiral emitters.

Supplementary Table 6. Summary of the calculated data for absolute configuration (R or P).

Molecule	R^a	g^a	\bar{R}^b	\bar{g}^b	g_{cal} reduction ^c	g_{PL} increment ^d
R_p -CzpPhTrz	-5.0×10^{-39}	-8.5×10^{-4}	-4.5×10^{-39}	-8.1×10^{-4}	4.7%	-
$(-)-(R,R)$ -CAI-Cz	1.7×10^{-39}	3.7×10^{-3}	1.5×10^{-39}	3.3×10^{-3}	10.8%	-
(R) -OBN-Cz	-7.6×10^{-40}	-6.6×10^{-4}	-5.0×10^{-40}	-4.6×10^{-4}	30.3%	-
(P) -Aza[6]helicene	-1.1×10^{-38}	-1.0×10^{-2}	-1.0×10^{-38}	-7.7×10^{-3}	23.0%	-
R-CzCN	1.6×10^{-39}	1.2×10^{-2}	7.8×10^{-40}	3.4×10^{-3}	71.7%	2.0-fold
(P,P)-CzTBCO	2.8×10^{-38}	1.5×10^{-2}	2.1×10^{-38}	1.0×10^{-2}	33.3%	1.3-fold

^aRotatory strengths (R , in units of esu^2cm^2), and g -factors calculated without considering vibrational effects using stationary-point TD-DFT calculations. ^bRotatory strengths (\bar{R} , in units of esu^2cm^2), and g -factors (\bar{g}) calculated with considering vibrational effects using nuclear ensemble approach. ^c g_{cal} reduction is defined as the reduction in g -factor due to molecular vibrations divided by the g -factor calculated without considering vibrational effects (from stationary-point TD-DFT calculations). ^d g_{PL} increment is defined as the enhancement factor of g_{PL} after deuteration.

Dear Editor and Reviewers,

Enclosed please find our revised manuscript entitled “**Deuteration promotes circularly polarized light emission by suppression of vibration**” (Manuscript ID: NCOMMS-24-50744C) for publication as a research article in *Nature Communications*. We sincerely thank the reviewers for their insightful and professional comments, which helped us substantially improve the quality of our manuscript.

We have meticulously revised the manuscript in response to Reviewer #3’s comments and the point-by-point responses are provided.

1. We have clarified that deuteration does not introduce steric effects. According to suggestion, we have revised the title from “**Deuteration promotes circularly polarized light emission**” to “**Deuteration promotes circularly polarized light emission by suppression of vibration**”.
2. We have provided theoretical calculation to address the issue of excitonic coupling.
3. We have incorporated additional CD and CPL measurements to validate our experimental results, and provided a reasonable explanation.

For your reference, annotated files of the revised manuscript and supporting information with all changes highlighted in yellow are also attached. We hope that the revised paper could now be satisfied for your requirements.

Yours Sincerely,

Chuluo Yang on behalf of the authors.

Corresponding author: **Professor Chuluo Yang**

at College of Materials Sciences and Engineering, Shenzhen University, Shenzhen 518071, People’s Republic of China. Tel: +86-0755-86713985, e-mail: clyang@szu.edu.cn.

Point-by-Point Response to Reviewers' Comments

Reviewer #2:

General comment: *All concerns have been addressed by the authors and I have no more comments.*

Response: We sincerely appreciate the reviewer for the positive comment.

Reviewer #3:

General comment 1: *The authors have attempted to address the concerns raised by the referees. While I acknowledged their revisions last time, I am now less convinced with their additional experimental work.*

Indeed, it now seems that the effect of the deuterium is more related to the dihedral angle within the biaryl electron acceptor bridge than to its involvement in the frontier molecular orbital. Accordingly, this is more of a steric effect that could eventually be obtained with another atom rather than a specific deuterium, if I am correct. In my opinion, the title should be changed in order to align more closely with the experimental results.

Response: We appreciate the reviewer for the critical comment, and regret any misunderstanding caused. We respectfully clarify that **deuteration does not introduce a steric effect, since deuterium and hydrogen have essentially identical van der Waals radius**. Accordingly, to capture deuteration's effects, we applied the Diagonal Born-Oppenheimer correction in our calculations. As the nuclear mass (μ) increases, the vibrational frequency ν decreases ($\nu = \frac{1}{2\pi} \frac{\sqrt{k}}{\sqrt{\mu}}$, *Nat. Commun.* **14**, 6481 (2023)), which in turn lowers the zero-point energy (*Nat. Photon.* **18**, 516-523 (2024)) and slightly shifts the vibrationally averaged nuclear coordinates (*Phys. Chem. Chem. Phys.* **22**, 24257-24269 (2020)). These subtle changes lead to minor variations in bond lengths, bond angles, and dihedral angles upon deuteration, as summarized in **Supplementary Table 5**. Importantly, these changes are thus intrinsic to isotopic substitution (specific to deuterium in this work) rather than steric interactions.

We have now explicitly presented these geometry variations in **Supplementary Table 5** for clarity. Regarding the title, taking into account of review's suggestion, we have revised it from "**Deuteration promotes circularly polarized light emission**" to "**Deuteration promotes circularly polarized light emission by suppression of vibration**". We have also updated the abstract accordingly.

Modified sentence, (manuscript, page 8): "...due to **slight shifts in the biphenyl backbone nuclear coordinates** upon deuteration (**Supplementary Table 5**)..."

Modified Abstract: “Such advancements are attributed to **suppression** of vibrations **by deuteration**.”

Supplementary Table 5. Comparison of bond lengths, bond angles, and dihedral angles between R-CzCN and R-D(16)CzCN.

	Atom1	Atom2	Atom3	Atom4	R-CzCN	R-D(16)CzCN
Bond #1	1	2	-	-	1.4410 Å	1.4366 Å
Bond #2	2	3	-	-	1.3909 Å	1.3896 Å
Bond #3	3	4	-	-	1.3955 Å	1.3954 Å
Bond #4	4	5	-	-	1.3929 Å	1.3931 Å
Bond #5	5	6	-	-	1.3924 Å	1.3928 Å
Bond #6	6	1	-	-	1.4425 Å	1.4394 Å
Bond #7	7	8	-	-	1.4471 Å	1.4454 Å
Bond #8	8	9	-	-	1.4096 Å	1.4094 Å
Bond #9	9	10	-	-	1.3744 Å	1.3744 Å
Bond #10	10	11	-	-	1.4130 Å	1.4140 Å
Bond #11	11	12	-	-	1.3815 Å	1.3803 Å
Bond #12	12	7	-	-	1.4381 Å	1.4341 Å
Angle #1	1	2	3	-	122.345°	122.497°
Angle #2	2	3	4	-	121.437°	121.128°
Angle #3	3	4	5	-	118.093°	118.251°
Angle #4	4	5	6	-	121.429°	121.422°
Angle #5	5	6	1	-	122.218°	122.003°
Angle #6	6	1	2	-	113.894°	114.194°
Angle #7	7	8	9	-	121.354°	121.060°
Angle #8	8	9	10	-	120.441°	120.474°
Angle #9	9	10	11	-	119.043°	119.212°
Angle #10	10	11	12	-	122.188°	121.832°
Angle #11	11	12	7	-	120.258°	120.435°
Angle #12	12	7	8	-	115.813°	116.113°
Dihedral #1	1	2	3	4	1.48°	1.23°
Dihedral #2	2	3	4	5	3.88°	3.67°
Dihedral #3	3	4	5	6	2.89°	2.61°
Dihedral #4	4	5	6	1	3.49°	3.37°
Dihedral #5	5	6	1	2	8.26°	7.73°

Dihedral #6	6	1	2	3	7.27°	6.70°
Dihedral #7	6	1	7	8	46.9°	47.5°
Dihedral #8	7	8	9	10	3.75°	3.81°
Dihedral #9	8	9	10	11	3.10°	3.00°
Dihedral #10	9	10	11	12	3.03°	3.08°
Dihedral #11	10	11	12	7	3.98°	3.74°
Dihedral #12	11	12	7	8	10.23°	10.02°
Dihedral #13	12	7	8	9	10.23°	10.14°

General comment 2: *The authors tried to respond to referee 1 but their response is far from being clear. They invoked an instrumental error in treating the data, and then used calculations to ascertain their “new” results. Based on their measurements, I am therefore also wondering if the effect of the deuterium may impact the potential presence of excitonic coupling within these chiral emitters. This would ultimately have a direct consequence on the chiroptical properties obtained in both the absorption and emission processes. Have the authors taken this into consideration?*

Response: We thank the reviewer for the valuable comments and apologize for the lack of clarity in our previous response to Referee 1 regarding the CD data. The issue indeed originated from a data processing error: we incorrectly used the g_{lum} formula to calculate g_{abs} . The g_{lum} formula is:

$$g_{lum} = \frac{2(A_{left} - A_{right})}{I_t}$$

where A_{left} and A_{right} are the absorption intensities of the left- and right-circularly polarized light, respectively, and I_t is the transmitted light intensity after passing through the sample. This error caused the inconsistency,

but it has now been corrected ($g_{abs} = \frac{2(A_{left} - A_{right})}{A_{left} + A_{right}}$).

Regarding the reviewer’s concern on the potential influence of deuteration on excitonic coupling within these chiral emitters: excitonic coupling arises from the interaction between the excited states (typically S_1 and S_2) and leads to energy-level splitting ($\Delta E_{S_2-S_1} = 2J$), where J is the excitonic coupling strength

(Comprehensive Chiroptical Spectroscopy. 1st ed. Weinheim: Wiley, 115-166 (2012); *Chem. Soc. Rev.* **36**, 914-931 (2007); *Chem. Eur. J.* **25**, 11294-11301 (2019); *Sci. China Chem.* (2025) <https://doi.org/10.1007/s11426-025-2716-6>). In our calculations, we explicitly applied the Diagonal Born-Oppenheimer correction to account for the increased nuclear mass of deuteration. As shown in **Supplementary Table 3**, deuteration leads to only minor shifts in the S₁ and S₂ energies; the S₁-S₂ gap and the corresponding excitonic coupling strength J show negligible variation. These results indicate that **deuteration does not affect excitonic coupling in our systems**.

Modified sentence, (manuscript, page 8): "...simulated CD spectra and **Supplementary Table 3 for excited states; deuteration does not affect excitonic coupling**)."

Supplementary Table 3. Energy levels of the S₁ and S₂ states, S₁-S₂ splitting and excitonic coupling strength for R-D(16)CzCN, and R-CzCN.

R-D(16)CzCN	Energy (eV)	S ₁	3.9132
		S ₂	3.9235
	Energy splitting (eV)	$\Delta E_{S_2-S_1}$	0.0103
	Excitonic coupling strength (eV)	J	0.0052
R-CzCN	Energy (eV)	S ₁	3.9157
		S ₂	3.9259
	Energy splitting (eV)	$\Delta E_{S_2-S_1}$	0.0102
	Excitonic coupling strength (eV)	J	0.0051

General comment 3: *I also have some concerns about the potential FRET mechanism. Why should there be a dependence on excitation wavelengths between 290 and 330 nm? The chiral donor should only transfer energy from its lowest excited state, if I am correct. Explanations based on ECD responses (Fig. S27, for example) are not relevant to me because the very low ECD values may lead to errors in interpretation.*

Response: We appreciate the reviewer's professional question. The excitation-wavelength dependence of the CPL spectra suggests that **two excitation pathways coexist**: (i) excitation of the chiral host (R/S-D(16)CzCN) followed by host→guest energy transfer; (ii) direct excitation of the achiral guest (BN2). As shown in Fig. 1b, the chiral host's absorption decreases sharply beyond 290 nm, whereas BN2 absorbs strongly in the range of 310-330 nm (*Adv. Funct. Mater.* **31**, 2102017 (2021)). Therefore, excitation at 310 or 330 nm increasingly

favors direct excitation of BN2 and reduces the fraction of excitations initiated on the chiral host; this should lower the contribution of host→guest energy transfer. This naturally leads to the observed decrease in CPL intensity.

To eliminate the possibility of measurement artefacts, we also carried out CD and CPL measurements in solution, where the chiral host and achiral guest (2 wt% BN2 in R/S-D(16)CzCN) were co-dissolved in THF/water mixtures ($V_{\text{THF}}:V_{\text{water}} = 10:90$) at a total concentration of 1 mM. This poor-solvent combination facilitates the formation of uniform nanoaggregates, providing an environment similar to that in thin films. A distinct CD signal was observed in the range of 400-500 nm, exceeding 1.5 mdeg (Supplementary Fig. 26d), confirming that the chiroptical response is genuine and not due to instrumental noise. Moreover, the CPL spectra of the same solution exhibited similar excitation-wavelength dependence (290-330 nm, Supplementary Figs. 26e,f), consistent with our thin-film CPL measurements.

Taken together, these results confirm that the excitation-wavelength dependence originates from the varying relative contributions of host→guest energy transfer and direct guest excitation, rather than from artefacts or misinterpretation of weak ECD signals.

Additional sentence, (Supplementary Information, Supplementary Note 3): “The CD spectra of R/S-D(16)CzCN:BN2 in THF/water mixtures ($V_{\text{THF}}:V_{\text{water}} = 10:90$) further confirm this signal.”

Supplementary Fig. 26. Chiroptical properties of R/S-D(16)CzCN:BN2. (a) Experimental CD spectra of

R/S-D(16)CzCN neat films and R/S-D(16)CzCN:BN2 films. **(b)** CP-PL spectra of R/S-D(16)CzCN:BN2 films under excitation at 290, 310, 330, 350, 370, 390 and 410 nm. **(c)** Column diagrams of $|g_{PL}|$ values of R/S-D(16)CzCN:BN2 films excited at 290, 310, 330, 350, 370, 390, 410 and 500 nm. **(d)** Experimental CD spectra of R/S-D(16)CzCN:BN2 in THF/water mixtures ($V_{THF}:V_{water} = 10:90$) at the concentration of 1 mM. **(e)** CP-PL spectra of R/S-D(16)CzCN:BN2 in THF/water mixtures ($V_{THF}:V_{water} = 10:90$) at the concentration of 1 mM, under excitation at 290, 310, and 330 nm. **(f)** Column diagrams of $|g_{PL}|$ values of R/S-D(16)CzCN:BN2 in THF/water mixtures ($V_{THF}:V_{water} = 10:90$) at the concentration of 1 mM, excited at 290, 310, and 330 nm.

We believe that these revisions fully address the reviewer's concerns and enhance the overall presentation of our work. We appreciate the reviewer's positive evaluation and valuable suggestions, which have greatly helped us refine our manuscript for publication.